# Active morphogenesis of patterned epithelial shells

**Diana Khoromskaia[1], Guillaume Salbreux[1,2]***

[1]The Francis Crick Institute, London, United Kingdom; [2]University of Geneva, Geneva, Switzerland

**Abstract** Shape transformations of epithelial tissues in three dimensions, which are crucial for embryonic development or in vitro organoid growth, can result from active forces generated within the cytoskeleton of the epithelial cells. How the interplay of local differential tensions with tissue geometry and with external forces results in tissue-scale morphogenesis remains an open question. Here, we describe epithelial sheets as active viscoelastic surfaces and study their deformation under patterned internal tensions and bending moments. In addition to isotropic effects, we take into account nematic alignment in the plane of the tissue, which gives rise to shape-dependent, anisotropic active tensions and bending moments. We present phase diagrams of the mechanical equilibrium shapes of pre-patterned closed shells and explore their dynamical deformations. Our results show that a combination of nematic alignment and gradients in internal tensions and bending moments is sufficient to reproduce basic building blocks of epithelial morphogenesis, including fold formation, budding, neck formation, flattening, and tubulation.

## Editor's evaluation

The article provides a physical description of shape transformations of epithelial tissues in three dimensions, subject to active forces generated within the cytoskeleton of the epithelial cells. The work is motivated by organoids and more generally by morphogenesis during development. Therefore, this study is useful not only for developmental biology but also for a general understanding of cellular properties, including membrane mechanics and cell shapes.

*For correspondence:
Guillaume.Salbreux@unige.ch

**Competing interest:** The authors declare that no competing interests exist.

## Introduction

Morphogenesis of embryos and the establishment of body shape rely on the three-dimensional deformation of epithelial sheets which undergo repeated events of expansion, contraction, convergence-extension, invagination, evagination, tubulation, and branching (*Gilbert and Barresi, 2020*). Tissue folding, for instance, is involved at different steps of embryogenesis (*Kominami and Takata, 2004*; *Sui et al., 2018*), organ (*Sumigray et al., 2018*), or entire organism development (*Livshits et al., 2017*; *Braun and Keren, 2018*). Recently, the growth of in vitro organoids, organ-like structures derived from stem cells capable of self-renewal and self-organisation, has revealed the intrinsic ability of biological systems to self-organise into complex structures from simple building blocks (*Huch et al., 2017*; *Kamm et al., 2018*; *Rossi et al., 2018*). Early steps in organoid self-organisation often start through the formation of a hollow, fluid-filled unpatterned sphere, undergoing spontaneous symmetry breaking (*Ishihara and Tanaka, 2018*) for example, in neural tube (*Meinhardt et al., 2014*) or intestinal (*Serra et al., 2019*; *Yang et al., 2021*) organoids. How this repertoire of shape changes and complex organisation emerges physically is a fundamental question.

Continuum theories of active materials, treating the epithelium as an active liquid crystal, have proven highly successful to achieve an understanding of the mechanics and flows of cellular collective

motion. Epithelia cultured in vitro exhibit patterns of orientational order and spontaneous flows which are consistent with predictions from hydrodynamic theories of active matter (*Duclos et al., 2017*; *Duclos et al., 2018*; *Blanch-Mercader et al., 2021a*). Constitutive equations involving a shear decomposition of tissue area and anisotropic elongation into cell shape changes, cell division, and cellular topological transitions can reproduce basic features of the developing *Drosophila* pupal wing (*Etournay et al., 2015*; *Popović et al., 2017*). Recently, several studies established a link between topological defects in tissue order, provided by cell elongation or internal anisotropic cellular structure, and morphogenetic events (*Kawaguchi et al., 2017*; *Saw et al., 2017*; *Mueller et al., 2019*; *Maroudas-Sacks et al., 2021*).

Here, we propose a description of three-dimensional deformations of a patterned epithelial spheroid, considered as a shell of active liquid crystal. We consider an active elastic shell theory which takes into account in-plane tensions and internal bending moments (*Lomholt, 2006*; *Maitra et al., 2014*; *Sahu et al., 2017*; *Salbreux and Jülicher, 2017*). Internal bending moments arise from an inhomogeneous distribution of stress across the tissue. Such inhomogeneities can arise from, for example, changes in cytoskeletal organisation along the epithelium apico-basal axis, or from apposed epithelial tissues with different mechanical properties (*Braun and Keren, 2018*; *Maroudas-Sacks et al., 2021*). Apico-basal gradients of contractility, for instance, play a key role in morphogenetic processes (*Martin and Goldstein, 2014*; *Sui et al., 2018*) and are effectively taken into account here by active bending moments.

We consider an initially spherically symmetric tissue subjected to spatially modulated internal forces. Our rationale is to consider a situation where chemical and mechanical processes are uncoupled, such that cell–cell communication mechanisms ensure symmetry-breaking of the sphere, which is then converted into a pattern of mechanical forces (*Ishihara and Tanaka, 2018*). We consider a particularly simple pattern where the spherical tissue is decomposed into two regions, subjected to different active forces, and explore shape changes that result from this pattern (*Figure 1a*). We compare the situation where internal tensions and bending moments are isotropic to a situation where a nematic field, provided by cellular anisotropic structures, orients the internal tensions and bending moments.

## Model
### Viscoelastic nematic active surface model for epithelial mechanics

We first discuss our mechanical description of the deforming tissue. We represent an epithelium as an active surface flowing with velocity $\mathbf{v}$ (*Salbreux and Jülicher, 2017*). The surface is taken to be elastic with respect to area changes, and fluid with respect to pure shear in the plane of the surface. Indeed, cellular rearrangements can fluidify in-plane epithelial flows by allowing cell elongation and cellular elastic stresses to relax on long time scales (*Popović et al., 2017*). Here, we consider such long enough time scales of hours to days which are relevant to organoid and developmental morphogenesis (*Gilbert and Barresi, 2020*). We also assume here that cell division and apoptosis or delamination are not occurring, such that elastic isotropic stresses do not relax (*Ranft et al., 2010*). Implicitly, we assume that cells have a preferred cell area.

Epithelia typically have a non-negligible thickness compared to characteristic transverse dimensions, and the apical and basal surfaces have different structures and are regulated differently. Notably, the basal surface is in contact with the basal lamina, a layer of extracellular matrix (*Khalilgharibi and Mao, 2021*). Therefore, a purely two-dimensional representation of epithelial stresses would miss essential aspects of their mechanics. We therefore introduce here the tension tensor $t^{ij}$, but also the bending moment tensor $m^{ij}$ which captures internal torques arising from differential stresses acting along the surface cross section (*Figure 1b and c*). We assume that the surface possesses a bending rigidity, captured by a bending modulus $\kappa$. When the curvature deviates from a flat layer, a bending moment results from the surface curvature (*Equation 6*). In addition, active bending moments can arise in the surface (*Salbreux and Jülicher, 2017*), for instance, due to actomyosin-generated differential active stresses along the apicobasal axis (*Messal et al., 2019*; *Fouchard et al., 2020*).

Cellular force generating elements are not necessarily isotropic; for instance, because cytoskeletal structures exhibit a preferred orientation (*Martin, 2020*) or inhomogeneous distribution across cellular interfaces (*Bertet et al., 2004*), or because the epithelial cells themselves exhibit an elongation axis

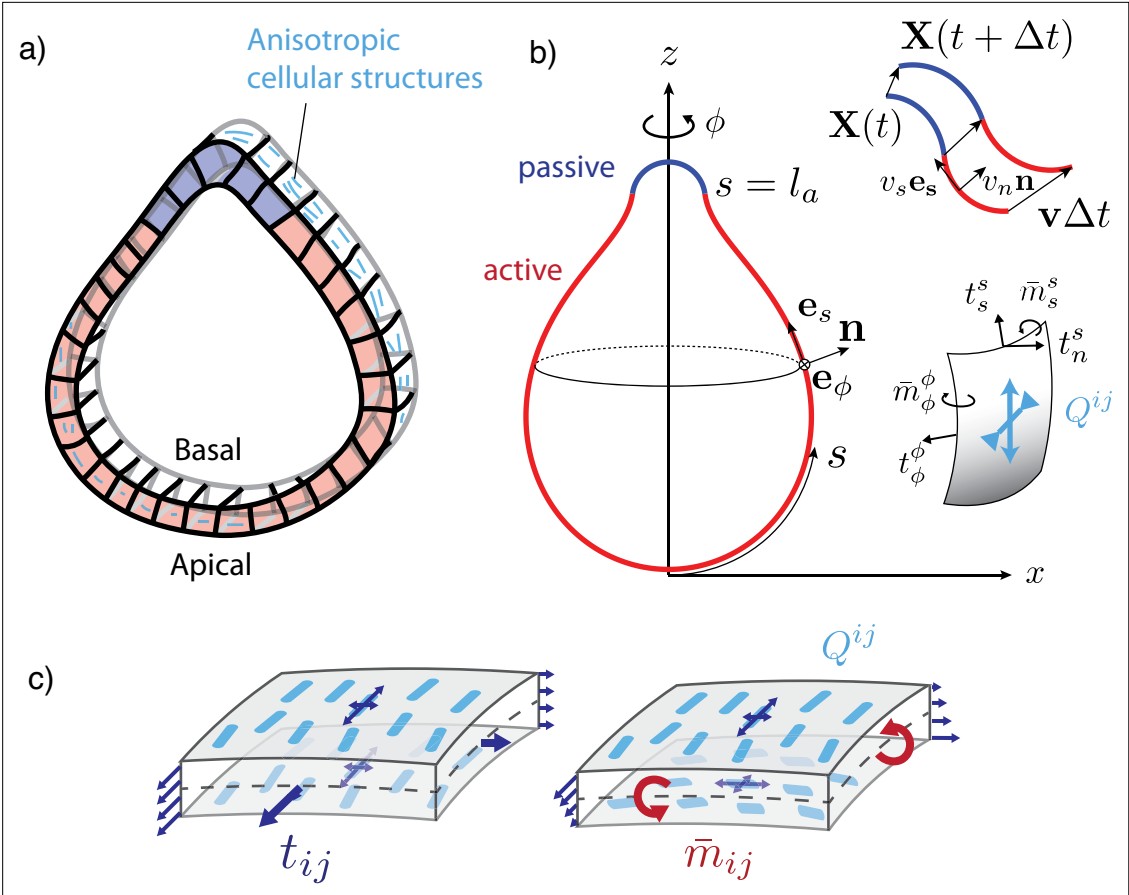

**Figure 1.** A two-dimensional surface with nematic order represents an epithelial sheet undergoing active deformations. (**a**) Schematic of an epithelial tissue with a cellular state pattern. (**b**) Parametrisation of the axially symmetric shell and its deformation with the flow $\mathbf{v}$, and components of the tension and torque tensors. We note that $m^{\phi s} = \bar{m}^{\phi\phi}x$ and $m^{s\phi} = -\bar{m}^{ss}/x$. (**c**) Stresses integrated across the thickness of the sheet result in tensions $t_{ij}$ and bending moments $m_{ij}$ acting on the midsurface. Anisotropic and possibly different tensions (dark-blue arrow crosses) on the apical and basal sides of the epithelium result in anisotropies in $t_{ij}$ and $\bar{m}_{ij}$, which can be captured by a nematic order parameter $Q_{ij}$ (e.g. blue rods on the top surface).

The online version of this article includes the following video for figure 1:

**Figure 1—video 1.** Deformation of an epithelial shell with free volume, $l_a/L_0 = 0.85$, $\delta\zeta_c = 15\kappa/R_0$, with active torque colour coded.
https://elifesciences.org/articles/75878/figures#fig1video1

**Figure 1—video 2.** Deformation of an epithelial shell with free volume, $l_a/L_0 = 0.85$, $\delta\zeta_c = 15\kappa/R_0$, with tangential velocity shown as white arrows and colour coded.
https://elifesciences.org/articles/75878/figures#fig1video2

**Figure 1—video 3.** Deformation of an epithelial shell with free volume, $l_a/L_0 = 0.35$, $\delta\zeta_c = 12.5\kappa/R_0$, with active torque colour coded.
https://elifesciences.org/articles/75878/figures#fig1video3

**Figure 1—video 4.** Deformation of an epithelial shell with free volume, $l_a/L_0 = 0.35$, $\delta\zeta_c = 12.5\kappa/R_0$, with tangential velocity shown as white arrows and colour coded.
https://elifesciences.org/articles/75878/figures#fig1video4

**Figure 1—video 5.** Deformation of an epithelial shell with conserved volume, $l_a/L_0 = 0.1$, $\delta\zeta_c = 40\kappa/R_0$, with active torque colour coded.
https://elifesciences.org/articles/75878/figures#fig1video5

**Figure 1—video 6.** Deformation of an epithelial shell with conserved volume, $l_a/L_0 = 0.1$, $\delta\zeta_c = 40\kappa/R_0$, with tangential velocity shown as white arrows and colour coded.
https://elifesciences.org/articles/75878/figures#fig1video6

**Figure 1—video 7.** Deformation of an epithelial shell with conserved volume, $l_a/L_0 = 0.7$, $\delta\zeta_c = 110\kappa/R_0$, with active torque colour coded.
https://elifesciences.org/articles/75878/figures#fig1video7

**Figure 1—video 8.** Deformation of an epithelial shell with conserved volume, $l_a/L_0 = 0.7$, $\delta\zeta_c = 110\kappa/R_0$, with tangential velocity shown as white arrows and colour coded.
https://elifesciences.org/articles/75878/figures#fig1video8

(*Duclos et al., 2017*). Therefore, we introduce a coarse-grained surface nematic order parameter $\mathbf{Q}$ which quantifies the average level of orientational order in the tissue. We assume that the nematic order parameter is tangent to the active surface.

## Force balance

On a curved surface we define the rotated bending moment tensor $\bar{m}^{ij} = -m^{ik}\epsilon_k{}^j$, which we adopt for convenience. The local force balance projected on the tangential and normal directions reads (*Salbreux and Jülicher, 2017*)

$$\nabla_i t^{ij} + C_i^j t_n^i = -f^{\text{ext},j} \tag{1}$$

$$\nabla_i t_n^i - C_{ij} t^{ij} = -f_n^{\text{ext}} - P, \tag{2}$$

where notations of differential geometry are introduced in Appendix 1; briefly $C_{ij}$ is the curvature tensor, $g_{ij}$ denotes the metric tensor, and $\epsilon_{ij}$ the antisymmetric Levi-Civita tensor, $\mathbf{n}$ the vector normal to the surface, $t^{ij}$ is the tangential contribution of the tension tensor and $t_n^i$ its normal contribution, and $\nabla_i$ denotes the covariant derivative on the surface. The tangential and normal torque balance provide the transverse tension and antisymmetric part of the tangent tension tensor:

$$t_n^i = \nabla_k \bar{m}^{ki}, \tag{3}$$

$$\epsilon_{ij} t^{ij} = C_{ij} m^{ij}. \tag{4}$$

We assume an external force density $\mathbf{f}^{\text{ext}} = f_n^{\text{ext}}\mathbf{n} + f^{\text{ext},j}\mathbf{e}_j$ acting on the surface in addition to a difference of hydrostatic (uniform) pressure $P = P_{in} - P_{out}$, but no external torques (*Figure 1*). Here, we consider situations at low Reynolds number, where inertial forces may be neglected, and where additional external forces are negligible, such that the surface as a whole is force-free, $\oint_S \text{d}S\, \mathbf{f}^{\text{ext}} = \mathbf{0}$. Dissipative couplings to the external fluid are ignored here as the characteristic viscosity of a biological tissue ($\sim 10^5$ Pa s; *Marmottant et al., 2009*; *Guevorkian et al., 2010*) is several orders of magnitude larger than that of water ($10^{-3}$ Pa s).

## Constitutive equations

In line with our hypothesis describing the material properties of an epithelium, we use the following constitutive equations:

$$t_s^{ij} = (2Ku + \zeta + (\eta_b - \eta)v_k^k)g^{ij} + 2\eta v^{ij} + \zeta_{\text{n}}Q^{ij}, \tag{5}$$

$$\bar{m}^{ij} = \left(2\kappa C_k^k + \zeta_c + \eta_{cb}\frac{\text{D}}{\text{Dt}}C_k^k\right)g^{ij} + \zeta_{cn}Q^{ij}. \tag{6}$$

where $t_s^{ij}$ is the symmetric part of the tension tensor and, on a curved surface, the strain rate tensor $v^{ij}$ and the corotational time derivative of the curvature tensor $\frac{\text{D}}{\text{Dt}}C^{ij}$ are given by (*Salbreux and Jülicher, 2017*)

$$v^{ij} = \frac{1}{2}(\nabla^i v^j + \nabla^j v^i) + C^{ij}v_n, \tag{7}$$

$$\frac{\text{D}}{\text{Dt}}C^{ij} = -\nabla^i(\partial^j v_n) - v_n C^i{}_k C^{kj} + v_k \nabla^k C^{ij} + \omega_n\left(\epsilon^{ik}C_k^j + \epsilon^{jk}C_k^i\right), \tag{8}$$

with $\omega_n = \frac{1}{2}\epsilon^{ij}\nabla_i v_j$ the normal component of the vorticity. $u$ is the area strain, measuring local changes of area relative to a reference value; a precise definition is introduced in *Equation 14*. $Q^{ij}$ is a traceless, symmetric tensor characterising nematic orientational order on the surface.

We now discuss these constitutive equations. The surface elastic response is determined by the area elastic modulus $K$ and the bending modulus $\kappa$. The dynamical deformations of the surface are characterised by the two-dimensional shear and bulk viscosities $\eta$ and $\eta_b$ and the bulk bending viscosity $\eta_{cb}$. While the shear and bulk viscosities penalise in-plane isotropic and anisotropic deformation rates, the bending viscosity penalises the rate of change of total surface curvature $C_k^k$. The bending viscosity dampens normal deformations and prevents bending modes, which would otherwise have no dissipative cost and could result in numerical instabilities.

The remaining contributions to *Equations 5; 6* proportional to $\zeta$, $\zeta_{\rm n}$, $\zeta_c$, $\zeta_{cn}$ correspond to active tensions and bending moments. $\zeta$ is an isotropic active surface tension, $\zeta_{\rm n}$ is the in-plane nematic active stress, with $\zeta_{\rm n} > 0$ usually referred to as the 'contractile' active stress and $\zeta_{\rm n} < 0$ as the 'extensile' active stress (*Marchetti et al., 2013*). $\zeta_c$ is the isotropic bending moment, which locally favours a spontaneous curvature $C_k^k = -\zeta_c/(2\kappa)$. If the active surface corresponds simply to two parallel layers under surface tension $\gamma_a$, $\gamma_b$ (such as an epithelium with apical surface tension $\gamma_a$ and basal surface tension $\gamma_b$), and separated by a distance $h$, an active isotropic bending moment $\zeta_c \sim h(\gamma_a - \gamma_b)/2$ emerges in the surface to lowest order in the curvature tensor. The term in $\zeta_{cn}$ corresponds to an anisotropic active bending moment. In the bilayer picture, where the active surface corresponds to two layers $a$ and $b$, it could generally arise from differences between the two layers in the level of order $Q_{ij}^a$ and $Q_{ij}^b$ or in the level of nematic active stress $\zeta_{\rm n}^a$ and $\zeta_{\rm n}^b$. For example, such differences could stem from two contractile (respectively extensile) layers with perpendicular nematic orientations $+Q_{ij}$ and $-Q_{ij}$ (*Figure 1c*), or from two layers with parallel nematic order, but one subjected to contractile active stresses and the other to extensile active stresses.

In the absence of external forces, deformations of the epithelial shell are driven by distributions of active tensions and bending moments, which are prescribed on it through the isotropic profiles $\zeta(s)$ and $\zeta_c(s)$, the anisotropic components proportional to $\zeta_{\rm n}(s)$ and $\zeta_{cn}(s)$, and the shape-dependent nematic order parameter.

We note that *Equations 5 and 6* can be seen as generic constitutive equations for a nematic active surface with broken up-down symmetry but no broken chiral or planar-chiral symmetry, arising from an expansion in the curvature tensor and in the nematic order parameter $Q_{ij}$ of the tensor $t_s^{ij}$ and $\bar{m}_{ij}$ (*Salbreux and Jülicher, 2017*; *Salbreux et al., 2022*). For simplicity some allowed additional couplings entering the generic constitutive equations have not been taken into account here, notably active contributions to the tension tensor (*Equation 5*) and bending moment tensor (*Equation 6*) proportional to the curvature tensor $C_{ij}$. *Salbreux et al., 2022* provide a more general list of possible couplings for active fluid nematic surfaces.

## Nematic order parameter

For simplicity here we assume that the nematic order parameter minimises an effective free energy, thus ignoring potential active effects on the ordering (*Salbreux et al., 2022*). We consider the following effective free energy of the nematic on a curved surface (*De Gennes and Prost, 1995*; *Jiang et al., 2007*; *Kralj et al., 2011*; *Pearce et al., 2019*):

$$F = \int \mathrm{dS} \left( \frac{k}{2} \left( \nabla_i Q^{jk} \right) \left( \nabla^i Q_{jk} \right) - \frac{a}{4} Q_{ij} Q^{ij} + \frac{a}{16} \left( Q_{ij} Q^{ij} \right)^2 \right), \tag{9}$$

with the Frank elastic constant $k$, which is assumed to be equal for all distortions. The Landau–de Gennes contribution is chosen such that for $k = 0$ the aligned state with $Q_{ij} Q^{ij} = 2$ is a minimiser for $a > 0$. Additional coupling terms between the nematic and curvature tensor are not considered here for simplicity (*Napoli and Vergori, 2012*).

# Deformations of a polarised active sphere

We now turn to describe axisymmetric deformations of a closed nematic active surface.

## Geometric setup

The epithelium is represented by a thin spherical shell undergoing axisymmetric deformations (*Figure 1b*). Its two-dimensional midsurface $\mathbf{X}(\phi, s) \in \mathbb{R}^3$ is parametrised by the arc length coordinate $s \in [0, L]$ and the angle of rotation $\phi \in [0, 2\pi]$ as

$$\mathbf{X}(\phi, s) = \left( x(s) \cos \phi, x(s) \sin \phi, z(s) \right). \tag{10}$$

The local tangent basis is given by $\{\mathbf{e}_\phi, \mathbf{e}_s\}$, and $\mathbf{n}$ is the outward-pointing surface normal. The geometry of axisymmetric surfaces is described further in Appendix 1. We require that the metric component $g_{ss} = 1$, which implies relations between the tangent angle $\psi(s) \in [0, \pi]$ and the shape functions $x(s)$ and $z(s)$

$$\partial_s x = \cos \psi, \tag{11}$$

$$\partial_s z = \sin \psi, \tag{12}$$

which, together with the meridional principal curvature

$$C_s^s = \partial_s \psi, \tag{13}$$

are sufficient to reconstruct the surface shape from the curvature $C_s^s$. In this axisymmetric setup, the velocity field reads $\mathbf{v} = v^s \mathbf{e}_s + v_n \mathbf{n}$, with $v^s$ the tangential and $v_n$ the normal velocities.

The undeformed initial surface is a sphere $S_0$ with radius $R_0$, and all quantities defined on it are denoted with a subscript '0'. We define the area strain on a point of the surface as

$$u = \frac{dS - dS_0}{dS_0}, \tag{14}$$

where $dS$ is the surface area element at the point considered on the surface, and $dS_0$ is the surface area element of the same material point on the sphere. With this definition, $u = 0$ on the initial sphere. We denote $s_0(s)$ the arc length position on the undeformed sphere $S_0$ of a material point at arc length position $s$ on the deformed sphere. One then has $u = f_\phi f_s - 1$ with $f_s = \frac{ds}{ds_0}$ the meridional stretch and $f_\phi = \frac{x}{x_0}$ the circumferential stretch. Integrating $f_s^{-1} = f_\phi/(u + 1)$ yields the arc length reparametrisation $s_0(s)$ between the initial and the deformed surface. The Lagrangian time derivative of the area strain (*Equation 14*) is related to the flow through

$$\frac{D}{Dt} u = (1 + u) v_k^k. \tag{15}$$

## Nematic order

Here, with axial symmetry, the nematic tensor $Q_{ij}$ has the non-zero component $q = Q_\phi^\phi = -Q_s^s$. On the closed shell, the nematic director (Appendix 3), which represents the alignment, will have two +1 topological defects at the poles (Figure 3a) as a consequence of the Poincaré–Hopf theorem (*Hopf, 1927*). The order parameter $q$ vanishes there, creating defect cores of size $l_c = \sqrt{k/a}$, which is the characteristic nematic length. In this geometry the Euler–Lagrange equation resulting from the free energy (*Equation 9*) is

$$\partial_s^2 q = \frac{1}{2l_c^2} q(q^2 - 1) + \frac{\cos \psi}{x} \left( 4 \frac{\cos \psi}{x} q - \partial_s q \right). \tag{16}$$

An example solution of *Equation 16* on the sphere is shown in Figure 3b. From the two possible states with $q = \pm 1$ in the bulk, respectively, we choose $q = 1$ for reference. This corresponds to circumferential alignment of the nematic order (Figure 3a, right). The sign of the tensions and bending moments is then only controlled by the $\zeta$-prefactors. For example, a nematic tension with $\zeta_n > 0$ corresponds to circumferential active contraction, resulting in an elongated shape. For nematic bending moments, if one chooses $Q_{ij}$ to represent the order parameter on the outer side of the shell, the sign convention is such that $\zeta_{cn} > 0$, $q > 0$ results in circumferential contraction on the outer side and contraction along the meridians on the inner side of the shell. We note that the shape is only influenced by the order parameter via the active tension $\zeta_n Q^{ij}$ and the active moment $\zeta_{cn} Q^{ij}$, but is otherwise insensitive to the nematic elastic energy (*Equation 9*). Minimisation of the Frank free energy by deformations of passive nematic surfaces has been previously discussed (*Jiang et al., 2007*).

## Active profiles

We consider initially spherical epithelial shells containing an active region that drives the deformation. For the steady-state analysis, this region is a circular patch of size $l_a \leq L_0$ (*Figure 1b*), such that the active terms are given on $S_0$ by step-like profiles, for example

$$\zeta_c(s_0) = \begin{cases} \zeta_c^0 + \delta\zeta_c, & \text{if } s_0 \in [0, l_a] \\ \zeta_c^0, & \text{otherwise} \end{cases} \tag{17}$$

and similarly for $\zeta(s_0)$, $\zeta_n(s_0)$, and $\zeta_{cn}(s_0)$. The circular patch deforms with the material points, which reflects that the active properties are associated with a predefined group of cells. If not stated otherwise, the values outside the active region are $\zeta^0 = \zeta_c^0 = \zeta_n^0 = \zeta_{cn}^0 = 0$. This passive part of the surface is governed by the constitutive *equations 5 and 6*, but with vanishing active terms.

In dynamical simulations, active tension and bending moment profiles are defined on the spherical surface at time $t = 0$ using sigmoid functions $f(x, \mu, \sigma)$ of the form

$$f(x, \mu, \sigma) = 1 - \left(1 + e^{-\frac{x-\mu}{\sigma}}\right)^{-1},$$ (18)

for their space and time dependence. For instance, the active bending moment profile is defined on $S_0$ as

$$\zeta_c(s_0, t = 0) = (1 - f(t = 0, \mu_t, \sigma_t))(\zeta_c^0 + \delta\zeta_c f(s_0, l_a, \sigma_s))$$ (19)

as a smooth version of the step-profile *Equation 17*, and $\zeta$, $\zeta_n$, and $\zeta_{cn}$ are defined analogously. The profile is then advected with the material points (*Figure 1b*), while its intensity increases through the time-dependent sigmoid (e.g. *Figure 2d*).

## Volume

We consider two possibilities for the volume enclosed by the epithelium. In one limit the tissue is assumed to be impermeable and the enclosed volume is treated as an incompressible fluid exerting hydrostatic pressure on the tissue. The volume is conserved when the shell deforms:

$$V = V_0,$$ (20)

with the pressure $P$ acting as the Lagrange multiplier.

In the other limit the tissue is fully permeable. At steady state, in this limit the volume can change freely and no pressure acts on the tissue, $P = 0$. In dynamical simulations, we introduce a volume viscosity $\eta_V$ such that the pressure is coupled to the volume change via

$$P = -\eta_V \partial_t V$$ (21)

where $\eta_V$ is a parameter chosen to be small enough that the internal pressure is small compared to other forces.

## Stationary shapes

For given profiles of active tensions and bending moments, steady-state shapes are obtained as solutions of the mechanical equilibrium equations. Those are a system of non-linear ode's containing the force and torque balances *Equations 1–4*, the geometric *Equations 11–13*, the constitutive relations *Equations 5–8* and *Equation 14* with vanishing velocities $v^s = v_n = 0$, and, if applicable, the nematic equilibrium *Equation 16*.

## Dynamical deformations

In the dynamical version of the model a given active profile generates a velocity $\mathbf{v}(\phi, s, t)$, whose normal part deforms the surface (*Figure 1b*). The components $\{v^s, v_n\}$ of this instantaneous velocity are obtained by solving the force and torque balance *Equations 1–4* (derived for the axisymmetric surface in *Equations 63–65*), together with the constitutive *Equations 5–8*, on the shape $\mathbf{X}(\phi, s, t)$. Since $u(s, t)$ and $Q_{ij}(s, t)$ are also given, these constitute a linear system of ode's. The shape is evolved in time in a Lagrangian approach, in which material points move according to the full-velocity vector $\mathbf{v}$,

$$\partial_t \mathbf{X} = \mathbf{v}.$$ (22)

Surface quantities, such as the active profiles and the area strain, are advected accordingly. The nematic order parameter evolves in time quasi-statically, where we assume that it relaxes instantaneously to the solution of *Equation 16* written on the deformed surface at time $t$.

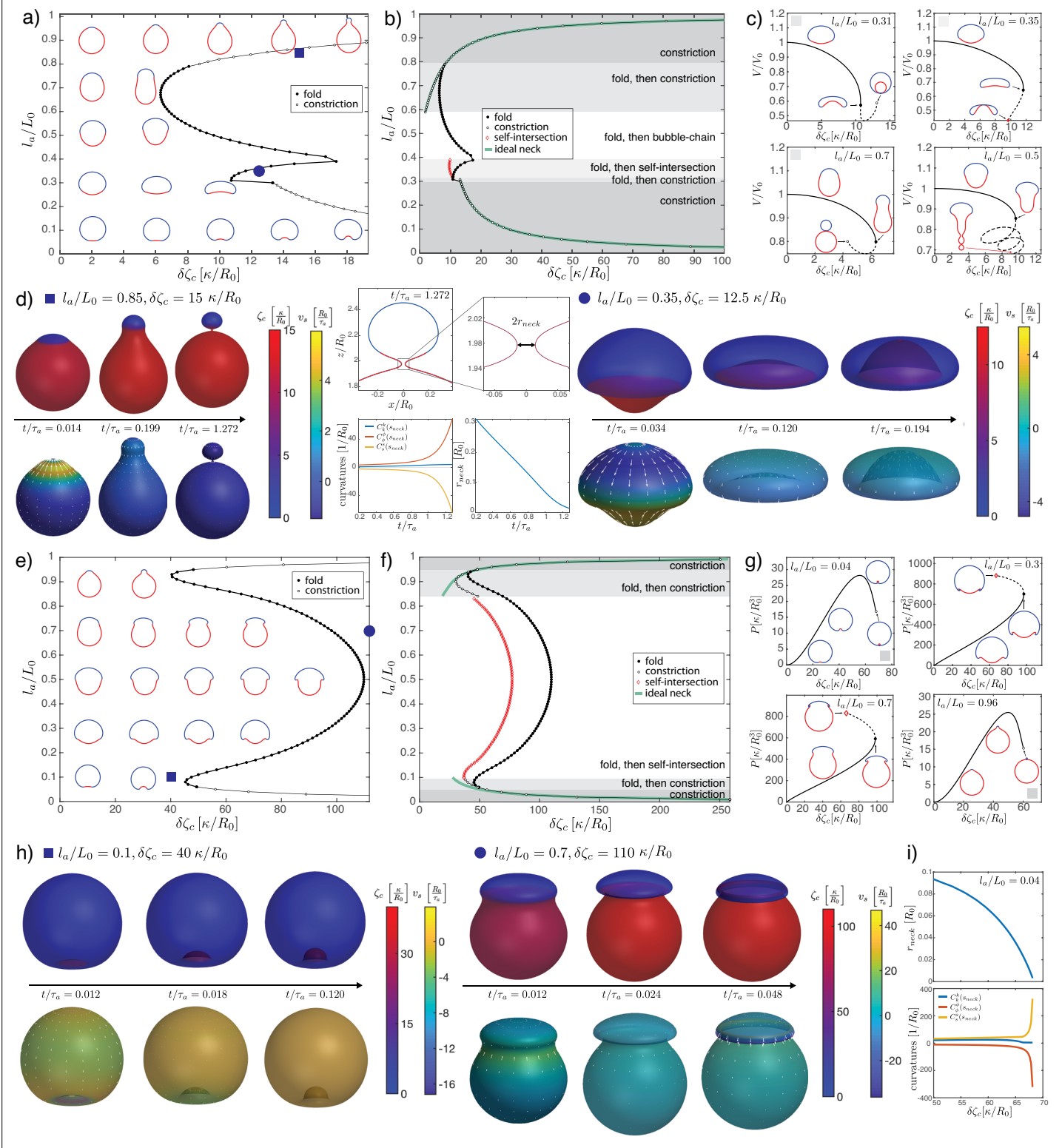

**Figure 2.** Deformations of epithelial shells due to active bending moments, with free (**a–d**) and conserved (**e–i**) volume. (**a, e**) Shape diagram. (**b, f**) Details of shape diagram illustrating different behaviours of solution branches. The ideal neck line (green) represents the bending moment difference required to create budded shapes consisting of two spheres with $u = 0$, as given by *Equation 24*. (**c**) Examples of solution branches in the ($\delta\zeta_c$, $V$)-plane corresponding to four different regions in (**b**). (**g**) Examples of solution branches in the ($\delta\zeta_c$, $P$)-plane chosen from three different regions

*Figure 2 continued on next page*

*Figure 2 continued*

in (**e**). (**d, h**) Dynamic simulations of shape changes, for parameter values indicated in the shape diagrams (**a, e**). (**i**) Neck radius and curvatures at the neck as functions of $\delta\zeta_c$ for the example $l_a/L_0 = 0.04$ in (**g**). Other parameters: $\tilde{K} = 10^3$, $\tilde{\eta}_{cb} = 10^{-2}$, $\tilde{\eta}_V = 10^{-4}$.

The online version of this article includes the following video and figure supplement(s) for figure 2:

**Figure 2—video 1.** Deformation of an epithelial shell with free volume, $l_a/L_0 = 1$, $\zeta_n = 1.5\kappa/R_0^2$, with nematic director shown with black lines and active nematic tension colour coded.

https://elifesciences.org/articles/75878/figures#fig2video1

**Figure 2—video 2.** Deformation of an epithelial shell with free volume, $l_a/L_0 = 1$, $\zeta_n = 1.5\kappa/R_0^2$, with tangential velocity shown as white arrows and colour coded.

https://elifesciences.org/articles/75878/figures#fig2video2

**Figure 2—video 3.** Deformation of an epithelial shell with free volume, $l_a/L_0 = 1$, $\zeta_n = 3\kappa/R_0^2$, with nematic director shown with black lines and active nematic tension colour coded.

https://elifesciences.org/articles/75878/figures#fig2video3

**Figure 2—video 4.** Deformation of an epithelial shell with free volume, $l_a/L_0 = 1$, $\zeta_n = 3\kappa/R_0^2$, with tangential velocity shown as white arrows and colour coded.

https://elifesciences.org/articles/75878/figures#fig2video4

**Figure 2—video 5.** Deformation of an epithelial shell with conserved volume, $l_a/L_0 = 0.3$, $\delta\zeta_n = 40\kappa/R_0^2$, with nematic director shown with black lines (where $\delta\zeta_n \neq 0$) and active nematic tension colour coded.

https://elifesciences.org/articles/75878/figures#fig2video5

**Figure 2—video 6.** Deformation of an epithelial shell with conserved volume, $l_a/L_0 = 0.3$, $\zeta_n = 40\kappa/R_0^2$, with tangential velocity shown as white arrows and colour coded.

https://elifesciences.org/articles/75878/figures#fig2video6

**Figure 2—video 7.** Deformation of an epithelial shell with conserved volume, $l_a/L_0 = 0.7$, $\delta\zeta_n = 60\kappa/R_0^2$, with nematic director shown with black lines (where $\delta\zeta_n \neq 0$) and active nematic tension colour coded.

https://elifesciences.org/articles/75878/figures#fig2video7

**Figure 2—video 8.** Deformation of an epithelial shell with conserved volume, $l_a/L_0 = 0.7$, $\zeta_n = 60\kappa/R_0^2$, with tangential velocity shown as white arrows and colour coded.

https://elifesciences.org/articles/75878/figures#fig2video8

**Figure supplement 1.** Details of the steady-state solutions with nearly closed necks formed by isotropic bending moments for free volume (**a**) and conserved volume (**b**), and $l_a/L_0 = 0.9$.

**Figure supplement 2.** Maximal relative surface area of the steady-state shapes measured along a solution branch for each $l_a/L_0$ in the case of conserved volume, corresponding to shapes shown in *Figure 2e–g*.

## Dimensionless variables

The equations are made dimensionless (marked by tilde) by rescaling tensions by $\kappa/R_0^2$, bending moment densities by $\kappa/R_0$, lengths by $R_0$, force densities by $\kappa/R_0^3$, viscosities by the two-dimensional shear viscosity $\eta$ of the epithelium, times by the characteristic time scale $\tau_a = \eta R_0^2/\kappa$, and velocities by $R_0/\tau_a$. This leaves the dimensionless parameters $\tilde{K} = KR_0^2/\kappa$, $\tilde{l}_c = l_c/R_0$, $\tilde{\eta}_b = \eta_b/\eta$, $\tilde{\eta}_{cb} = \eta_{cb}R_0^2/\eta$ and $\tilde{\eta}_V = \eta_V R_0^4/\eta$ to be fixed. We choose to set $\tilde{\eta} = \tilde{\eta}_b = 1$, $\tilde{\eta}_V = 10^{-4}$ for fast relaxation of the volume, and the nematic length scale is set to $\tilde{l}_c = 0.1$. Working under the assumptions of linear shell theory for a homogeneous thin shell (*Reddy, 2006*), one can relate the elastic moduli to each other via the thickness $h$ of the cell layer, and express $\tilde{K} = 12(R_0/h)^2$. In simulations we use $\tilde{K} = 1000$, corresponding to $h/R_0 \approx 0.1$, which covers a range of systems from gastrulating embryos (e.g. sea urchin *Davidson et al., 1995*) to organoids (*Serra et al., 2019*). Similarly, for the bulk bending viscosity we have $\tilde{\eta}_{cb} \sim (h/R_0)^2 = 10^{-2}$.

## Numerical methods

For both the steady-state computation and the dynamics, the resulting sets of ode's are integrated numerically with the boundary-value-problem solver bvp4c of MATLAB, which implements a fourth-order collocation method on an adaptive spatial grid (*Kierzenka and Shampine, 2001*). The equations are solved on the full interval $[0, L]$, and geometrical singularities at the poles are handled using analytical limits at $s = 0, L$ (Appendix 6). Any integral constraint, such as volume conservation, is rewritten as a boundary value problem and added to the system of ode's to be solved.

The dynamics simulations start with a sphere at time $\tilde{t} = 0$. We study each of the four active effects separately. The corresponding active profile is switched on smoothly via a sigmoid function in time, such that it reaches its target intensity at $\tilde{t} \approx 0.02$. The time integration according to *Equation 22* is done with an explicit Euler method with adaptive step size via

$$\mathbf{X}'(\phi, s, t + \delta t) = \mathbf{X}(\phi, s, t) + \delta t \mathbf{v}(\phi, s, t). \tag{23}$$

In order to keep the force and torque balance equations in the form given by *Equations 63–65*, the updated surface is reparametrised as $\mathbf{X}'(\phi, s', t + \delta t)$ in a new arc length $s'(s)$ which is calculated from the condition $g_{s's'} = 1$. The profiles and surface quantities are passed between time steps as spline interpolants.

To produce the diagrams of steady-state shapes, $l_a$ is fixed and the control parameter is the difference of the active profile value between the passive and the active regions of the shell, for example, for the profile given in *Equation 17* it is $\delta\zeta_c$. A solution branch is found by starting from the spherical solution at zero difference of active profile, and calculating a sequence of steady-state shapes, progressively increasing the magnitude of the difference in activity. Two different methods are used to construct the solution branch for a sequence of control parameter values. For small values, starting from zero, the solution branch is obtained by making small increments in the control parameter. For larger values we switch to an implicit stepping method, which we developed based on a parametric representation of the solution branch (see Appendix 6 section 'Construction of solution branches'). This second method allows us to continue the solution branches into regions where the steady-state shapes become non-unique in the control parameter.

Details of the numerical methods can be found in Appendices 6 and 7 for the steady state and the dynamics simulations, respectively.

## Results

### Epithelia as active membranes: Isotropic active tensions

We first consider deformations of an epithelial shell due to patterns of isotropic active tensions and bending moments. A spatially varying isotropic tension represents a change in the preferred area of the epithelium due to either changes in sheet thickness or cell number (*Popović et al., 2017*). However, one can show that a step-profile of positive (contractile) tension $\zeta > 0$ does not lead, at steady state, to a three-dimensional deformation of the shell away from a spherical shape, which is a consequence of the absence of shear elasticity in our model (Appendix 8). Instead, the epithelium remains spherical and regions with higher tension contract. This leads to a rescaling of the relative active region size $l_a/L_0$ and, if the volume is free to change, also to a decrease in shell radius (Appendix 8). If the tension becomes negative, a buckling of the surface may occur (*Salbreux and Jülicher, 2017*). Here, we focus on positive tensions; therefore, if only isotropic active effects are considered, active internal bending moments are required to drive deformations away from the spherical shape.

### Epithelia as active shells: Isotropic active bending moments

We now turn to deformations induced by an increasing active bending moment in a spherical cap. In *Figure 2a and e*, we plot a phase diagram of steady-state shapes as a function of the increased active bending moment $\delta\zeta_c$ and the size of the active region $l_a$. The steady-state deformed shapes are plotted with the active region shown in red and the 'passive' region, where $\zeta_c = 0$, shown in blue. We can contrast the situation where fluid is free to exchange across the surface and at steady state the difference of pressure across the surface vanishes, $P = 0$ (*Figure 2a–d*), to the case where the volume enclosed by the surface is constrained to a fixed value (*Figure 2e–i*).

An isotropic active bending moment (term in $\zeta_c$ in *Equation 6*) induces a preferred curvature $(C^0)^k_k = -\frac{\zeta_c}{2\kappa}$, such that regions of a spherical shell with $\zeta_c > 0$ can be expected to flatten or bend inwards. Specifically, a difference of $\delta\zeta_c$ applied at the boundary of the active cap induces a jump in meridional curvature $C^s_s$ and a local folding of the sheet. Due to the spherical topology, the shape of the whole shell is affected by this fold, as can be seen from the sequences of stationary shapes obtained by increasing $\delta\zeta_c$ for intermediate values of $l_a/L_0$ (*Figure 2a*). In particular, for the same value of $\delta\zeta_c$ the active region may bend inward or keep a positive curvature, depending on its size.

When $l_a/L_0$ is small or close to 1, the resulting shape is characterised by the formation of a bud which form either inwards ($l_a \ll L_0$) or outwards ($L_0 - l_a \ll L_0$). In these cases, for sufficiently large values of $\delta\zeta_c$ the steady-state solution is lost through the formation of a constricting neck. In our simulations the constricting neck is numerically resolved up to values of $\sim 10^{-3} R_0$; extrapolation indicates full constriction at a finite $\delta\zeta_c$ (*Figure 2i*). As the neck radius decreases the principal curvatures at the neck diverge as $C_s^s, C_\phi^\phi \to \pm\infty$, such that $C_k^k$ remains finite (*Figure 2i*) and therefore the limiting, budded shape is a true steady-state solution. Such a transition is reminiscent of models of lipid membrane vesicles, which can be induced to form a budded shape consisting of two spheres connected by an infinitesimal region called the ideal neck (*Seifert et al., 1991*; *Jülicher and Lipowsky, 1993*; *Fourcade et al., 1994*; *Jülicher and Lipowsky, 1996*; *Seifert, 1997*). For lipid membranes the ideal neck condition gives the difference in spontaneous curvature between the two domains at which a vesicle will form two spheres, $1/R_1 + 1/R_2 = C_0$ with $R_1$ and $R_2$ the radius of the two spheres and $C_0$ the spontaneous curvature (*Seifert, 1997*). Here the choice of constitutive *Equations 5 and 6* does not correspond to the Helfrich model, and we find alternative matching conditions for the two regions connected by the infinitesimal neck: we find that $t_s^s$ changes sign across the neck, while $\bar{m}_s^s$ is continuous. This result can be derived by a scaling analysis around the neck (Appendix 2). In the free volume case, these conditions are satisfied when the active and passive regions are separated by the neck, and have the shapes of spheres with vanishing strain ($u = 0$) and radii $R_a$, $R_p$, related by the condition:

$$
\begin{aligned}
\frac{1}{R_a} - \frac{1}{R_p} &= -\frac{\delta\zeta_c}{4\kappa}, \\
-\frac{1}{R_a} - \frac{1}{R_p} &= -\frac{\delta\zeta_c}{4\kappa},
\end{aligned}
\tag{24}
$$

where the change of sign in the second line arises because the active region deforms inward and form a sphere with a negative mean curvature. The additional condition of vanishing strain $u = 0$ gives an additional relation for $R_1$ and $R_2$ as a function of $l_a/L_0$. Combining these conditions determine a curve in the parameter space $\delta\zeta_c R_0/\kappa$, $l_a/L_0$, which matches with the numerically determined curve of neck constriction (*Figure 2b*). In the fixed volume case, the matching conditions do not result in such a simple shape solution; however, using the same condition as for the free volume case appears to still provide a good approximation of the constriction point for small ($l_a \ll L_0$) and close to $L_0$ ($L_0 - l_a \ll L_0$) values of $l_a$ (*Figure 2f*). We conclude that infinitesimal neck formation can arise outside of the Helfrich model and that the ideal neck condition which is satisfied there does not generally extend to other models of surface mechanics.

At sufficiently large increase in the active bending moment difference $\delta\zeta_c$ and for intermediate values of $l_a/L_0$, a fold in the solution branch in the ($\delta\zeta_c$, $V$)-plane appears (*Figure 2c*). For most values of $l_a/L_0$, this fold is associated to the loss of a continuously attainable solution with increasing $\delta\zeta_c$, and a shape transition (*Figure 2b and c*). We expect shapes obtained by following the continuous branch of shapes beyond the fold to be unstable (Appendix 9). The (potentially unstable) physical branch eventually stops either through a self-intersection of the sheet at the poles (*Figure 2c*, $l_a/L_0 = 0.35$) or through the constriction of a small neck that develops near the boundary of the passive and active regions and separates the shell into two smaller, approximately spherical compartments (*Figure 2c*, $l_a/L_0 = 0.31, 0.7$). Alternatively the solution branch continues in a sequence of loops and the active region elongates (*Figure 2c*, $l_a/L_0 = 0.5$), forming an increasing number of bubble-like compartments.

Since we follow continuous trajectories of steady-state shapes in parameter space, we cannot directly obtain alternative steady-state solution branches after the shape transition. Therefore, we turn to dynamic simulations where we explicitly calculate flow fields, starting from the reference spherical shape, and evolve the surface shape (*Figure 2d*) with parameters chosen to be away from the transition in parameter space (*Figure 2a*). This also allows to resolve the sequence of shapes and velocity fields leading to a given steady-state deformed shape (*Figure 2h*, $l_a/L_0 = 0.1, \delta\zeta_c = 40\kappa/R_0$). For parameters beyond the shape transition, we find that a small neck can form, separating roughly the active and passive regions, whose radius decreases to 0 over time (*Figure 2d*). Alternatively the surface ends up self-intersecting (*Figure 2d*, $l_a/L_0 = 0.35, \delta\zeta_c = 12.5\kappa/R_0$). We do not find therefore alternative solution branches beyond the shape instability. Since intersection of the surface with itself is described by different physical interactions than considered here, our framework does not answer what would happen beyond the self-intersection line. However, assuming that self-intersection results in fusion

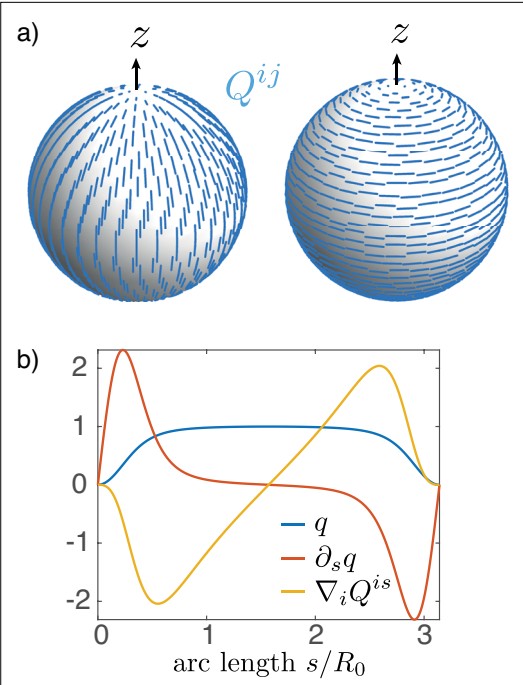

**Figure 3.** Nematic order on a sphere. (**a**) Two possible configurations for the nematic order parameter $Q_{ij}$ on a sphere with a + 1 topological defect at each pole: meridional (left) or circumferential (right) alignment. The order parameter minimises an effective energy (**Equation 9** with $l_c = 0.1R_0$). (**b**) Order parameter $q(s) = Q_\phi^\phi(s)$ as a solution of the Euler–Lagrange **Equation 16** on a sphere with $R_0 = 1$ and $l_c = 0.1R_0$; $q = 1$ at the equator and $q = 0$ at the locations of the defects (poles). For uniform $\zeta_n$, $\zeta_n \nabla_i Q^{is}$ is the active nematic contribution to the tangential force balance (**Equation 63**) and, close to the equator, results in the elongation of the surface along the axis of symmetry for $\zeta_n > 0$, and its contraction for $\zeta_n < 0$.

and rupture of the apposed two surfaces, active isotropic bending moment difference could in principle drive a change in tissue topology, from one sphere to two ($l_a/L_0 = 0.85, \delta\zeta_c = 15\kappa/R_0$), or from a sphere to a torus via self-intersection ($l_a/L_0 = 0.35, \delta\zeta_c = 12.5\kappa/R_0$).

When volume is conserved, deformations are broadly similar but tend to be more localised to the fold at the active boundary (**Figure 2e–i**). For intermediate values of $l_a/L_0$, the shell deforms into locally folded shapes, which eventually self-intersect at large bending moment difference (**Figure 2g**, $l_a/L_0 = 0.3, 0.7$, **Figure 2h**).

## Nematic active tensions

We now introduce the nematic order parameter $Q_{ij}$ and consider shape changes driven by contractile or extensile active stress in the active region (**Figure 3**). As expected, solving for the nematic order parameter profile on the undeformed sphere results in maximal order at the equator and two defects at the poles where the nematic order parameter vanishes, $q = 0$ (**Figure 3**). Two solutions with $q < 0$ and $q > 0$ can exist; in the following we take the convention that $Q_\phi^\phi = q > 0$, $Q_s^s = -q < 0$, corresponding to circumferential alignment of the order parameter, such that a contractile active stress ($\zeta_n > 0$) results in a positive circumferential tension, $t_\phi^\phi > 0$. Due to invariance of the constitutive equation by exchange $Q_{ij} \to -Q_{ij}$, $\zeta_n \to -\zeta_n$, the same shape deformations occur when considering meridional alignment of the order parameter ($q < 0$) and exchanging contractile ($\zeta_n > 0$) and extensile ($\zeta_n < 0$) active stresses.

As before, we study the cases of vanishing pressure difference across the shell (**Figure 4a–e**) and constrained volume inside the shell (**Figure 4f–i**). With a nematic tension profile on the surface, a deformation away from the spherical shape occurs even for homogeneous active nematic tension, $l_a/L_0 = 1$ (**Figure 4a–e**).

In the extensile case $\zeta_n < 0$ (or in the contractile case $\zeta_n > 0$ if $q < 0$), and no pressure difference across the shell, the surface progressively flattens into a flat, double-layered disc (**Figure 4b**, $l_a/L_0 = 1$, $\zeta_n < 0$). There is no shape transition occurring; instead, we find that the shape converges to a limit shape as $|\zeta_n| \to \infty$ (Appendix 4). The limit shape corresponds to two parallel flat discs of radius $R_d$, separated by a distance $2h$, connected by a narrow curved region. An asymptotic analysis (Appendix 4) shows that the radius of the disc and the separating distance obey the scaling relations, in the limit $\kappa \ll Kl_c^2$:

$$R_d \sim l_c, \quad h \sim \left(\frac{\kappa l_c}{K}\right)^{\frac{1}{3}} \tag{25}$$

The first relation shows that the limit shape has the size of the characteristic nematic length $l_c$. Physically, for $l_c \ll L_0$, the nematic active tension results in a contraction of the shape, until the shape is sufficiently close to the defect core for the nematic order to 'dissolve,' thus limiting further increase in the active tension.

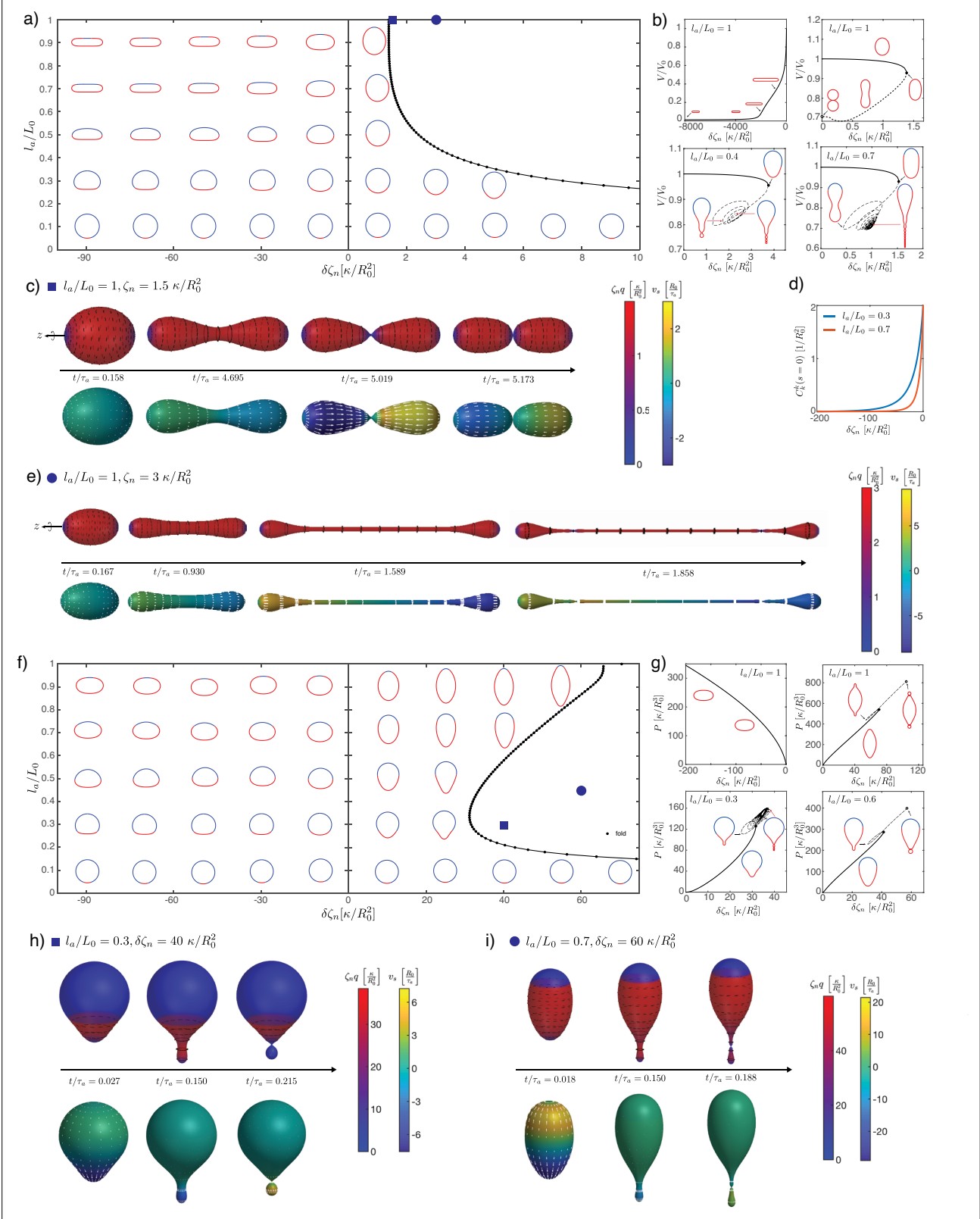

**Figure 4.** Deformations of epithelial shells due to nematic tensions, with free (**a–e**) and conserved (**f–i**) volume. (**a, e**) Shape diagrams. (**b, g**) Details of shape diagram illustrating the behaviour of solution branches. (**d**) Curvature at the south pole for extensile stress. (**c, e, h, i**) Dynamic simulations of shell shape changes, for parameter values indicated in the phase diagrams (**a, f**). Other parameters: $\tilde{K} = 10^3$, $\tilde{\eta}_{cb} = 10^{-2}$, $\tilde{\eta}_V = 10^{-4}$, $\tilde{l}_c = 0.1$.

*Figure 4 continued*

The online version of this article includes the following video and figure supplement(s) for figure 4:

**Figure 4—video 1.** Deformation of an epithelial shell with free volume, $l_a/L_0 = 1$, $\zeta_{cn} = -5\kappa/R_0$, with nematic director (black lines, as described in figure caption) and active nematic torque colour coded.

https://elifesciences.org/articles/75878/figures#fig4video1

**Figure 4—video 2.** Deformation of an epithelial shell with free volume, $l_a/L_0 = 1$, $\zeta_{cn} = -5\kappa/R_0$, with tangential velocity shown as white arrows and colour coded.

https://elifesciences.org/articles/75878/figures#fig4video2

**Figure 4—video 3.** Deformation of an epithelial shell with free volume, $l_a/L_0 = 0.3$, $\delta\zeta_{cn} = -20\kappa/R_0$, with nematic director (black lines, as described in figure caption) and active nematic torque colour coded.

https://elifesciences.org/articles/75878/figures#fig4video3

**Figure 4—video 4.** Deformation of an epithelial shell with free volume, $l_a/L_0 = 0.3$, $\delta\zeta_{cn} = -20\kappa/R_0$, with tangential velocity shown as white arrows and colour coded.

https://elifesciences.org/articles/75878/figures#fig4video4

**Figure 4—video 5.** Deformation of an epithelial shell with free volume, $l_a/L_0 = 1$, $\zeta_{cn} = 4\kappa/R_0$, with nematic director (black lines, as described in figure caption) and active nematic torque colour coded.

https://elifesciences.org/articles/75878/figures#fig4video5

**Figure 4—video 6.** Deformation of an epithelial shell with free volume, $l_a/L_0 = 1$, $\zeta_{cn} = 4\kappa/R_0$, with tangential velocity shown as white arrows and colour coded.

https://elifesciences.org/articles/75878/figures#fig4video6

**Figure 4—video 7.** Deformation of an epithelial shell with free volume, $l_a/L_0 = 0.3$, $\delta\zeta_{cn} = 15\kappa/R_0$, with nematic director (black lines, as described in figure caption) and active nematic torque colour coded.

https://elifesciences.org/articles/75878/figures#fig4video7

**Figure 4—video 8.** Deformation of an epithelial shell with free volume, $l_a/L_0 = 0.3$, $\delta\zeta_{cn} = 15\kappa/R_0$, with tangential velocity shown as white arrows and colour coded.

https://elifesciences.org/articles/75878/figures#fig4video8

**Figure 4—video 9.** Deformation of an epithelial shell with conserved volume, $l_a/L_0 = 0.3$, $\delta\zeta_{cn} = -150\kappa/R_0$, with nematic director (black lines, as described in figure caption) and active nematic torque colour coded.

https://elifesciences.org/articles/75878/figures#fig4video9

**Figure 4—video 10.** Deformation of an epithelial shell with conserved volume, $l_a/L_0 = 0.3$, $\delta\zeta_{cn} = -150\kappa/R_0$, with tangential velocity shown as white arrows and colour coded.

https://elifesciences.org/articles/75878/figures#fig4video10

**Figure 4—video 11.** Deformation of an epithelial shell with conserved volume, $l_a/L_0 = 0.5$, $\delta\zeta_{cn} = 50\kappa/R_0$, with nematic director (black lines, as described in figure caption) and active nematic torque colour coded.

https://elifesciences.org/articles/75878/figures#fig4video11

**Figure 4—video 12.** Deformation of an epithelial shell with conserved volume, $l_a/L_0 = 0.5$, $\delta\zeta_{cn} = 50\kappa/R_0$, with tangential velocity shown as white arrows and colour coded.

https://elifesciences.org/articles/75878/figures#fig4video12

**Figure supplement 1.** Details of dynamics simulations for shells with (**a, b**) homogeneous and (**c**) patterned nematic tension, which result in one or two constricting necks.

In the contractile case ($\zeta_n > 0$), the shape elongates until a shape transition is reached, characterised by a fold in the solution branch (*Figure 4b*, $l_a/L_0 = 1$, $\zeta_n > 0$). Following the solution branch after the fold eventually gives rise to a sequence of presumably unstable shapes with the formation of a central constricting neck. Intrigued by this result, we performed dynamical simulations for contractile active tensions above the shape transition (*Figure 4c and e*; *Figure 4—figure supplement 1*). Dynamic simulations show separation of the shape into two or more compartments via dynamical neck constrictions, with the neck radius vanishing over time (*Figure 4—figure supplement 1a*). Within the neck, $q \to 0$ as a result of the diverging principal curvatures (as can be seen from the presence of a term $(\frac{\cos(\psi)}{x} q)^2$ term in the nematic free energy, *Equation 102*). In particular, for values close to the branch fold (*Figure 4c*) the dynamics is reminiscent of cell division; however, in contrast to existing models of cell division (*Salbreux et al., 2009*; *Turlier et al., 2014*), the constriction appearing here does not require a narrow peak of active stress around the equator to occur. At larger contractile stress (*Figure 4e*), a narrow, elongated tube forms around the equator. This tube thins out over time,

and two symmetric necks emerge and constrict, suggesting that the shape would eventually separate into three topologically separated surfaces (*Figure 4—figure supplement 1b*).

For $0 < l_a/L_0 < 1$ and extensile stress in the active region $\delta\zeta_n < 0$, the active region tends to flatten more and more strongly as $|\delta\zeta_n|$ is increased, and the total curvature vanishes at the south pole ($C_k^k \to 0$, *Figure 4d*). For $0 < l_a/L_0 < 1$ and contractile stress $\delta\zeta_n > 0$, a fold in the solution branch appears at large value of $\delta\zeta_n$ (*Figure 4b and d*). Following the solution branch beyond the fold results in a complex trajectory in parameter space, corresponding to successive additions of new bubbles to a linear chain of bubbles within the active region. This bubble chain is observed both with free or constrained volume (*Figure 4b and g*). Here, we cannot conclude however whether these shapes are unstable. Instead, we consider the shape dynamics for $\delta\zeta_n$ values larger than the shape transition, here at fixed internal volume (*Figure 4h and i*). Here, a neck forms within the active region and its constriction leads to the separation of a smaller bubble. For small enough $l_a$ the smaller bubble appears nematic-free and spherical (*Figure 4h*, *Figure 4—figure supplement 1b*). This is consistent with restoration of isotropic state stability which can occur on a sphere whose size becomes smaller or comparable to $l_c$ (Appendix 3 section 'Stability of the isotropic state on a sphere').

## Active nematic bending moments

We now turn to shape deformations resulting from active bending moments oriented along the nematic order $Q_{ij}$. As for nematic tension, we adopt the convention of nematic alignment along the circumference, $Q_\phi^\phi = q > 0$; alignment along the meridians can be studied simply by changing the sign of the active coefficient $\zeta_{cn}$.

We first discuss the case where the nematic active bending moment is homogeneous ($l_a/L_0 = 1$), where there is no difference of pressure across the surface, and where $\zeta_{cn} = \delta\zeta_{cn} < 0$ (*Figure 5a–c and g*). We find that the sphere deforms into a shape with a central cylindrical part (*Figure 5a and b*). The length of the cylindrical part increases with increasing value of $|\zeta_{cn}|$. To characterise this, we note that the corresponding steady-state shape solutions have vanishing tensions $t_s^s = 0$ and $t_n^s = 0$ everywhere (*Figure 5—figure supplement 1*) and the force balances *Equations 63 and 64* are trivially satisfied. The torque balance *Equation 65* reads

$$2\kappa\partial_s C_k^k - \zeta_{cn}\partial_s q = 2\zeta_{cn}\frac{\cos\psi}{x}q. \tag{26}$$

Combining *Equations 26 and 48* one obtains that $\mathcal{L}[C_s^s - C_\phi^\phi - \zeta_{cn}q/(2\kappa)] = 0$, with the operator $\mathcal{L} = \partial_s + 2\frac{\cos\psi}{x}$. Solutions to $\mathcal{L}[f] = 0$ have the form $f = A/x^2$ with A a constant. The boundary condition that the function $f$ should be finite at the poles requires $A = 0$, such that

$$C_s^s - C_\phi^\phi = \frac{q\zeta_{cn}}{2\kappa}. \tag{27}$$

As a result, if the shape has a cylindrical part, in which $C_s^s = 0$ and $q = 1$, then the cylinder radius $R_c$ is given by

$$\frac{1}{R_c} = -\frac{\zeta_{cn}}{2\kappa}, \tag{28}$$

and since such solutions are area-preserving, with $u = 0$, the length of the cylindrical part scales as $L_c \sim 1/R_c$. These relations are in excellent agreement with simulation results for large enough $|\zeta_{cn}|$ (*Figure 5g*).

When $l_a < L_0$, the active region forms an outward cylindrical protrusion (*Figure 5a, b and g*) whose radius is still well described by *Equation 28*, replacing $\zeta_{cn}$ by $\delta\zeta_{cn}$, the value of the active nematic bending moment in the active region (*Figure 5g*). Using that within the cylindrical protrusion $u = 0$ so that the cylindrical protrusion has the same area as the original active domain and the relation *Equation 85* for the size of the active domain, we find that the length of the active protrusion is now given by

$$L_c \simeq \frac{R_0^2}{R_c}\left(1 - \cos\frac{l_a}{R_0}\right) = -\frac{\delta\zeta_{cn}R_0^2}{2\kappa}\left(1 - \cos\frac{l_a}{R_0}\right), \tag{29}$$

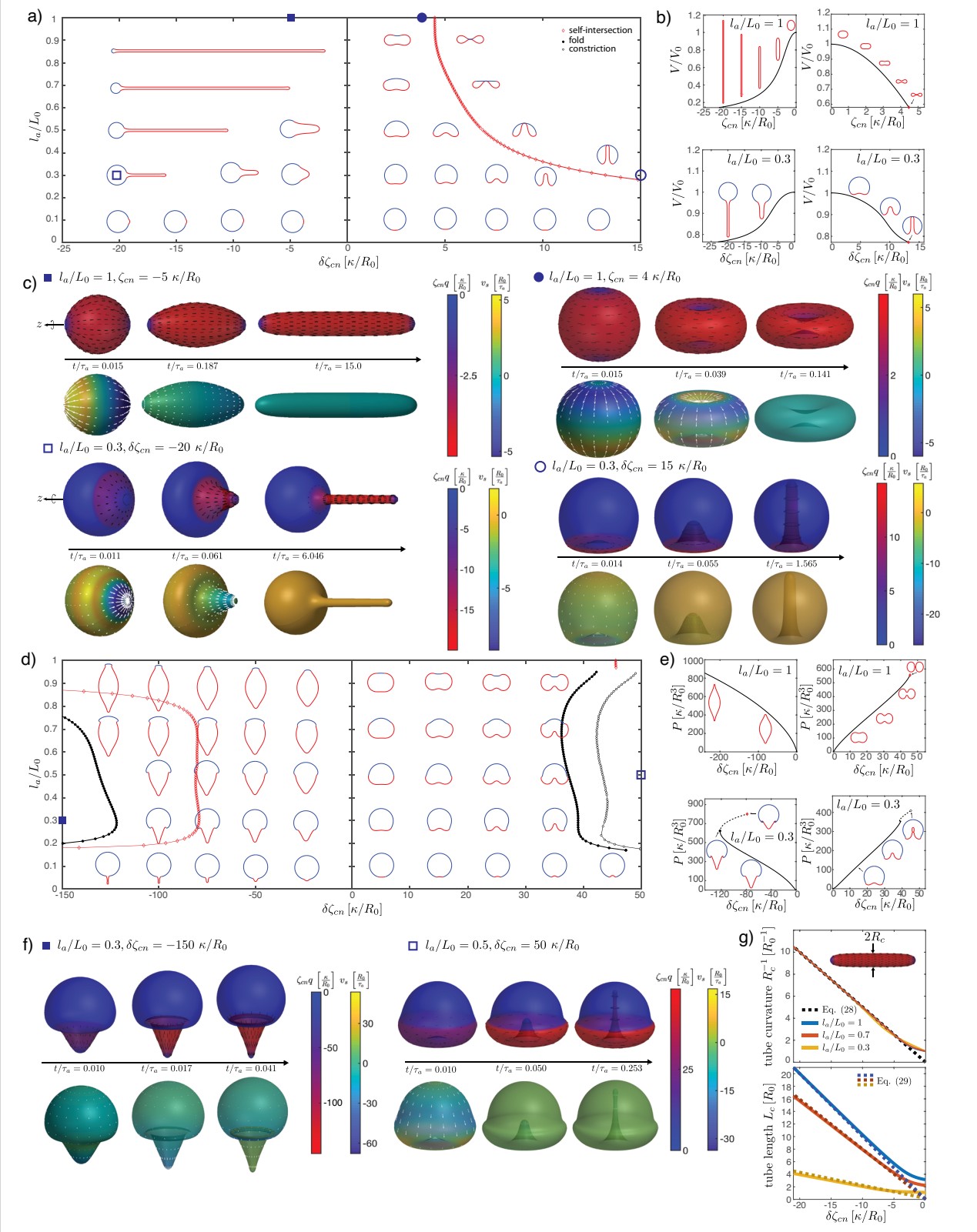

**Figure 5.** Deformations of epithelial shells due to nematic bending moments, with free (**a–c**) and conserved (**d, e**) volume. (**a, d**) Shape diagrams. (**b, e**) Details of shape diagram illustrating the behaviour of solution branches. (**c, f**) Dynamic simulations of shell shape changes, for parameter values indicated in the phase diagrams (**a, d**). In both cases in (**f**) the dynamics results in self-intersection. (**g**) Comparison of curvature and length of the cylindrical tubes for $l_a/L_0 = 1, 0.7, 0.3$, $\delta\zeta_{cn} < 0$ with analytical predictions. The tube length is measured on the steady-state shape as the arc length of

*Figure 5 continued on next page*

*Figure 5 continued*

the deformed active region, $s_{tube} = s(s_0 = l_a)$, and the tube curvature as $C_\phi^\phi(s_{tube}/2)$. Other parameters: $\tilde{K} = 1000$, $\tilde{\eta}_{cb} = 10^{-2}$, $\tilde{\eta}_V = 10^{-4}$, $\tilde{l}_c = 0.1$. In (c), (f), for $\delta\zeta_{cn}, \zeta_{cn} < 0$ the orientation of the director field drawn on the surface (black lines) is set by $-Q_{ij}$.

The online version of this article includes the following figure supplement(s) for figure 5:

**Figure supplement 1.** Details of steady-state shapes resulting from nematic bending moments with $\zeta_{cn} < 0$ and free volume.

which is again in excellent agreement with numerical simulation for large $|\delta\zeta_{cn}|$ and for different values of $l_a/L_0$ (**Figure 5g**).

For $\zeta_{cn} > 0$ and $l_a/L_0 = 1$ we find erythrocyte-like shapes, where the indentations at the poles become stronger with $\zeta_{cn}$ until the two poles touch (**Figure 5b**). This behaviour remains for $l_a < L_0$, resulting in a self-intersection line in the phase diagram (**Figure 5a**). Here, the shape can take the form of an inner tube entering the spherical shell (**Figure 5b**), reminiscent of epithelial shape changes observed during sea urchin gastrulation (**Ettensohn, 1984**).

Interestingly, when $l_a/L_0 < 1$ and the volume is free to change, both signs of $\delta\zeta_{cn}$ result in a cylindrical appendage forming from the active region. The sign of $\delta\zeta_{cn}$ determines whether the cylinder forms outside or inside of the remaining, roughly spherical shape. Dynamics simulations confirm that the shapes described above are stable solutions (**Figure 5c**). At the tip of the emerging cylinder lies the +1 topological defect. For $\delta\zeta_{cn} < 0$, when the protrusion grows towards the outside, such a situation is reminiscent of the observation of nematic defects in *Hydra*, where a set of topological defects, with +1 defects at the tip, have been observed in growing tentacles (**Maroudas-Sacks et al., 2021**). There, actin layers are perpendicular to each other, with circumferential alignment in the inner cell layer and longitudinal in the outer layer, which would indeed result in $\delta\zeta_{cn} < 0$ with our sign convention if the layers are contractile.

We now describe surfaces with fixed volume (**Figure 5d–f**). Here, we do not observe cylindrical shapes or protrusions as in the case of free volume. When $\zeta_{cn} < 0$ and $l_a = L_0$ the surface becomes spindle-like, narrowing at the poles with increasing $|\zeta_{cn}|$. As in the free volume case, when $\zeta_{cn} > 0$ the two opposite poles come in contact with each other (**Figure 5e**); such that subsequent fusion of the poles would lead to an overall toroidal shape of the shell. The shapes become more complex for $l_a < L_0$. Shape transitions occur at large $|\delta\zeta_{cn}|$, for both $\delta\zeta_{cn} < 0$ and $\delta\zeta_{cn} > 0$ (**Figure 5e**). In the case $\delta\zeta_{cn} < 0$, for increasing magnitude of the active bending moment, the shape becomes increasingly

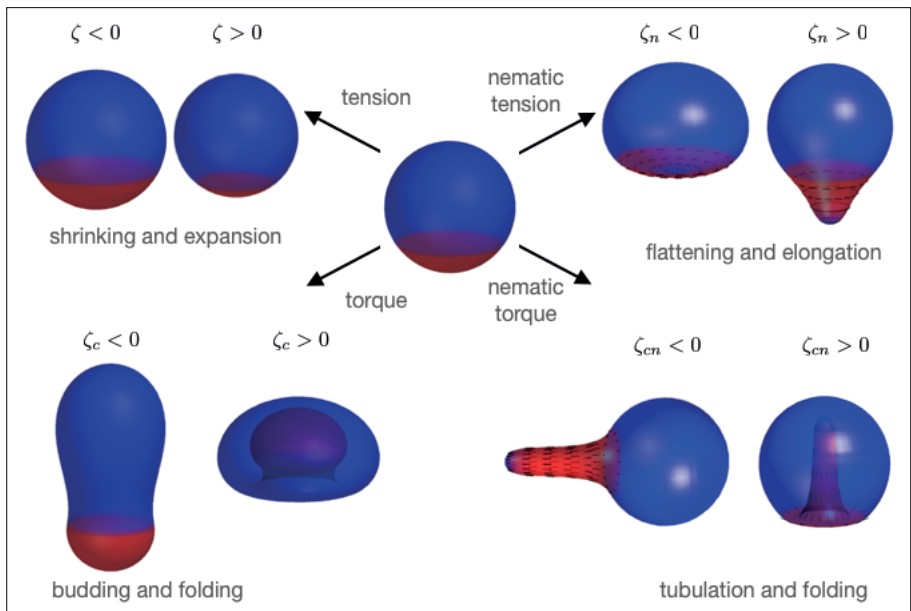

**Figure 6.** Summary of shape changes obtained through patterning of isotropic and anisotropic active tensions and bending moments. Active tensions and bending moments are present only in the red region of the surface. For $\zeta_{cn} < 0$ the director field orientation (black lines) is set by $-Q_{ij}$.

curved at the boundary between the passive and active regions, until the solution is lost. In the case $\delta\zeta_{cn} > 0$, the shell indents within the active region and the solution branch has a fold. To the right of the fold line in the shape diagram, the steady-state solutions are eventually lost through the formation of a small neck that separates off a smaller, internalised compartment. In contrast to the case of isotropic bending moments, here the sign of $\delta\zeta_{cn}$ determines whether the active region folds inwards or outwards, independent of the initial size $l_a/L_0$. As before, we use dynamics simulations to study the deformations for large $|\delta\zeta_{cn}|$ (*Figure 5f*). For both signs of $\delta\zeta_{cn}$, these result in shapes that are self-intersecting either along a circle ($l_a/L_0 = 0.3$, $\delta\zeta_{cn} = -150\kappa/R_0$) or at the poles ($l_a/L_0 = 0.5$, $\delta\zeta_{cn} = 50\kappa/R_0$).

## Discussion

In this study of deformations of patterned nematic active surfaces, we have found a diverse zoology of possible shape changes (*Figure 6*), characterised by budding and neck constrictions, transition of sphere to cylinder, tubulation, and flattening. We find that introduction of a nematic field on the surface greatly increases the space of possible shapes. Overall our work contributes to the characterisation of the 'morphospace' which biological systems can explore.

Some of our findings recapitulate epithelial deformations observed in biological systems. The flattening observed for an extensile homogeneous nematic surface (*Figure 4b*, $l_a/L_0 = 1$) could in principle lead to merging of the two apposed surfaces into a double-layer for large $|\zeta_n|$. Such a process of tissue planarisation appears to occur as an intermediate step in skin organoid formation, where epithelial cysts fuse and merge to form transient bilaterally symmetric structures (*Lei et al., 2017*). The formation of tubular appendages from nematic bending moments appears to recapitulate growth/regeneration of elongated bodies and tentacles in *Hydra* (*Maroudas-Sacks et al., 2021*) and, with an opposite sign, of epithelial invagination during sea urchin embryo gastrulation (*Ettensohn, 1984*).

The axisymmetric structure we have considered here naturally gives rise to two +1 nematic defects at the poles (*Figure 3a*). These defects then structure the nematic field and, as a result, the shape changes driven by nematic active tension or bending moments. Such an interplay between topological defect and shape changes is a recurring theme that may play a key role in morphogenesis (*Frank and Kardar, 2008*; *Metselaar et al., 2019*; *Hoffmann et al., 2021*; *Blanch-Mercader et al., 2021a*; *Blanch-Mercader et al., 2021b*). In practice +1 nematic defects are unstable to separation into two +1/2 defects; however, it is conceivable that a polar or additional weakly polar field stabilises the +1 defects (*Amiri et al., 2022*). Extension of the present work beyond axisymmetric structures will allow to distinguish more clearly the purely nematic and polar cases.

Continuum theories for curved surfaces, such as the Helfrich theory, have been extremely successful to describe shape transformations of passive vesicles, including homogeneous or phase-separated vesicles with coexisting domains (*Seifert et al., 1991*; *MacKintosh and Lubensky, 1991*; *Jülicher and Lipowsky, 1993*; *Seifert, 1997*; *Allain et al., 2004*; *Sens and Turner, 2004*; *Bassereau et al., 2014*). The effect of broken symmetry variables on passive surfaces, arising, for instance, from molecular tilt giving rise to polar order on a lipid membrane, has been considered theoretically (*MacKintosh and Lubensky, 1991*; *Lubensky and Prost, 1992*; *Park et al., 1992*). Continuum theories of active surfaces can similarly allow to study epithelial deformations (*Salbreux and Jülicher, 2017*; *Morris and Rao, 2019*; *Messal et al., 2019*). We note some important differences between the active surface model described here and passive membranes. (i) Our constitutive equations for tensions and bending moments *Equations 5 and 6* do not in general derive from a free energy (*Salbreux and Jülicher, 2017*) and describe a system out-of-equilibrium; (ii) while lipid membranes are nearly incompressible and are usually treated as surfaces with constant area, cells within epithelial tissues can change their area significantly (*Latorre et al., 2018*), which prompted us to consider a finite area modulus $K$: for example, simulations with constant volume have relative area changes of up to 20% (*Figure 2—figure supplement 2*); (iii) patterns of active tensions and bending moments imposed here also do not derive from an energy and are thought to respond to spatiotemporal chemical cues: in contrast, phase-separated domains in passive lipid vesicles obey equilibrium thermodynamics and their size is controlled, for instance, by line tension at the domain boundary (*Jülicher and Lipowsky, 1993*). In some cases, however, a similarity appears between shape transformations obtained in the active model we study here and the passive Helfrich model. For instance, budding occurring in lipid membranes due to phase separation of domains with different spontaneous curvature (*Jülicher and*

*Lipowsky, 1993*) is similar to the budding we observe here for different regions with different active isotropic bending moments.

We find here that nematically oriented active bending moments can give rise to spontaneous cylindrical tubes, without external force application (*Figure 5*). Spontaneous formation of hollow cylindrical vesicles with polar order due to molecular tilt has been discussed *Lubensky and Prost, 1992*; there the cylindrical shapes are considered to be open and the gain in defect energy allows the open cylinder to be more stable than the spherical shape. In contrast, we find here active surfaces which spontaneously form tubes, but stay closed and keep their topological charge. It has also been reported that a supported bilayer membrane under compression can spontaneously form tubes under negative tension (*Staykova et al., 2013*). In this work we have chosen to consider only positive isotropic tension; negative isotropic tension could give rise to further buckling instabilities. Models for chiral lipid bilayers in a tilted fluid phase have also predicted tubular shapes (*Helfrich and Prost, 1988*; *Selinger and Schnur, 1993*; *Selinger et al., 1996*; *Tu and Seifert, 2007*). Here, we have not considered chiral effects. These effects could be introduced by generalising the constitutive *Equations 5 and 6*, including terms which appear for surfaces with broken planar-chiral or chiral symmetry (*Salbreux and Jülicher, 2017*).

In contrast to purely elastic models of morphogenesis (*Höhn et al., 2015*; *Haas et al., 2018*), we have considered here morphogenetic events occurring on time scales long enough for shear elastic stresses to be relaxed by cell topological rearrangements, such that the tissue exhibits fluid behaviour (*Popović et al., 2017*). Whether a tissue behaves as an elastic or fluid material on time scales relevant to morphogenesis can in principle be probed experimentally (*Mongera et al., 2018*).

While we have focused the interpretation of our results to epithelial mechanics, the constitutive *Equations 5 and 6* we have considered here are generic and may also describe the large-scale behaviour of active nematics formed with cytoskeletal filaments and motors on a deformable surface (*Keber et al., 2014*). We considered here, however, a situation where the two-dimensional fluid has area elasticity, whereas cytoskeletal networks can in principle be fluid with respect to both shear and bulk shear due to the turnover of components.

In this study, we have considered chemical and mechanical processes to be uncoupled, except for the profile of active tension or torque being advected with the surface flow. Introducing additional couplings explicitly in this framework will extend the repertoire of shapes considered here. A natural choice is to consider the effect of a chemical undergoing reaction-diffusion on the surface and advected by the fluid, regulating active forces on the surface (*Mietke et al., 2019a*; *Mietke et al., 2019b*). Here, we assumed that orientational order relaxes quickly compared to other dynamical processes; in future work, this assumption could be lifted and one could study in particular how chemical regulation could influence the dynamics of orientational order in the tissue. Cells could also be sensing their own curvature and actively adapt their behaviour accordingly (*Chen et al., 2019*), which could lead to a dependency of the active coupling coefficients $\zeta$, $\zeta_n$, $\zeta_c$ or $\zeta_{cn}$ on the trace or determinant of the curvature tensor $C_{ij}$. It would be interesting to explore shapes arising from such a feedback. Volume conservation at cellular level could also be included explicitly, for instance, by introducing a tissue height field (*Morris and Rao, 2019*). Finally, we have considered here a tissue with a fixed preferred area, implicitly assuming that the epithelium is not growing. Tissue growth is a key aspect of biological development (*Gokhale and Shingleton, 2015*; *Eder et al., 2017*), and cell division and death can fluidify elastic stresses in an epithelium (*Ranft et al., 2010*); adding regulated growth in the model will be a step forward in our understanding of active morphogenesis of biological tissues.

## Acknowledgements

We thank S Grigolon for useful discussions and N Cuny for comments on the manuscript. DK and GS acknowledge support from the Francis Crick Institute, which receives its core funding from Cancer Research UK (FC001317), the UK Medical Research Council (FC001317), and the Wellcome Trust (FC001317), and from the CRUK multidisciplinary project award C55977/A23342.

## Additional information

### Funding

| Funder | Grant reference number | Author |
|---|---|---|
| Cancer Research UK | C55977/A23342 | Guillaume Salbreux |
| Cancer Research UK | FC001317 | Guillaume Salbreux |
| Medical Research Council | FC001317 | Guillaume Salbreux |
| Wellcome Trust | FC001317 | Guillaume Salbreux |

The funders had no role in study design, data collection and interpretation, or the decision to submit the work for publication. For the purpose of Open Access, the authors have applied a CC BY public copyright license to any Author Accepted Manuscript version arising from this submission.

### Author contributions
Diana Khoromskaia, Conceptualization, Software, Investigation, Visualization, Methodology, Writing – original draft; Guillaume Salbreux, Conceptualization, Supervision, Funding acquisition, Investigation, Methodology, Writing – original draft

### Author ORCIDs
Diana Khoromskaia ⓘ http://orcid.org/0000-0003-2597-6336
Guillaume Salbreux ⓘ http://orcid.org/0000-0001-7041-1292

### Decision letter and Author response
Decision letter https://doi.org/10.7554/eLife.75878.sa1
Author response https://doi.org/10.7554/eLife.75878.sa2

## Additional files

### Supplementary files
• Transparent reporting form

### Data availability
The current manuscript is a computational study, so no data have been generated for this manuscript. Modelling code is available at the GitHub repository https://github.com/DianaKhoromskaia/EpithelialShell, (copy archived at swh:1:rev:0838f09f1b2228d8d7da5183fc68f3b49c6ee734).

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

## Appendix 1

## Differential geometry of axisymmetric surfaces

### General relations of differential geometry

#### Fundamental tensors

A general framework for the mechanics of active curved surfaces is given in *Salbreux and Jülicher, 2017*; *Salbreux et al., 2022*, and we follow the differential geometry notation introduced there. Let $\mathbf{X} = \mathbf{X}(s^1, s^2)$ be a curved surface embedded in $\mathbb{R}^3$ and parametrised by the generalised coordinates $s^1, s^2$. A local covariant basis in the tangent plane is given by the vectors $\mathbf{e}_i = \partial_i \mathbf{X} = \partial \mathbf{X}/\partial s^i$ and the unit normal vector is constructed as $\mathbf{n} = (\mathbf{e}_1 \times \mathbf{e}_2)/|\mathbf{e}_1 \times \mathbf{e}_2|$ and chosen to point outwards for a closed surface in our convention. These define the metric tensor $g_{ij} = \mathbf{e}_i \cdot \mathbf{e}_j$ and the curvature tensor $C_{ij} = -(\partial_i \mathbf{e}_j) \cdot \mathbf{n} = \mathbf{e}_i \cdot \partial_j \mathbf{n}$. The infinitesimal surface and line elements are given by $\mathrm{dS} = \sqrt{g}ds^1 ds^2$ and $dl^2 = g_{ij} ds^i ds^j$, where $g$ is the determinant of the metric tensor. The antisymmetric Levi–Civita tensor is defined as $\epsilon_{ij} = \mathbf{n} \cdot (\mathbf{e}_i \times \mathbf{e}_j)$.

#### Covariant derivatives

The Christoffel symbols of the second kind are obtained from derivatives of the metric as

$$\Gamma_{ij}^k = \frac{1}{2} g^{km} \left( \partial_j g_{im} + \partial_i g_{jm} - \partial_m g_{ij} \right),$$

(30)

and Christoffel symbols of the first kind are defined as $\Gamma_{kij} = \frac{1}{2} \left( \partial_j g_{ki} + \partial_i g_{kj} - \partial_k g_{ij} \right)$. The covariant derivatives of a tangent vector field $f^i$ and a tangent tensor field $T^{ij}$, respectively, are given by

$$\nabla_i f^j = \partial_i f^j + \Gamma_{ik}^j f^k,$$

(31)

$$\nabla_i T^{jk} = \partial_i T^{jk} + \Gamma_{il}^j T^{lk} + \Gamma_{il}^k T^{jl}$$

(32)

In the following, we also use the divergence theorem on curved surfaces for a tangent vector field $\mathbf{f}$ (*Salbreux and Jülicher, 2017*),

$$\int_{\mathcal{S}} \mathrm{dS} \nabla_i f^i = \int_{\mathcal{C}} dl \nu_i f^i.$$

(33)

### Infinitesimal variation of surface quantities

For a small deformation of the surface

$$\delta \mathbf{X} = \delta X^i \mathbf{e}_i + \delta X_n \mathbf{n},$$

(34)

the variations of the basis vectors, the normal vector, the metric, and the mixed curvature tensor components are given by (*Salbreux and Jülicher, 2017*)

$$\delta \mathbf{e}_i = \left( \nabla_i \delta X^j + C_i^{\ j} \delta X_n \right) \mathbf{e}_j + \left( \partial_i \delta X_n - C_{ij} \delta X^j \right) \mathbf{n},$$

(35)

$$\delta \mathbf{n} = \left( -\partial_i \delta X_n + C_{ij} \delta X^j \right) \mathbf{e}^i,$$

(36)

$$\delta g_{ij} = \nabla_i \delta X_j + \nabla_j \delta X_i + 2 C_{ij} \delta X_n,$$

(37)

$$\frac{\delta \sqrt{g}}{\sqrt{g}} = \frac{1}{2} g^{ij} \delta g_{ij} = \nabla^k \delta X_k + C_k^{\ k} \delta X_n,$$

(38)

$$\delta C_i^{\ j} = -\nabla_i \left( \partial^j \delta X_n \right) + \left( \nabla_i \delta X^k \right) C_k^{\ j} - \left( \nabla^k \delta X^j \right) C_{ik} + \left( \nabla_k C_i^{\ j} \right) \delta X^k - \delta X_n C_{ik} C^{kj}.$$

(39)

These relations can be used to obtain time derivatives of surface quantities using $\delta \mathbf{X} = \delta t \mathbf{v}$ (Lagrangian surface update) or $\delta \mathbf{X} = \delta t v_n \mathbf{n}$ (where the surface shape is updated with the normal flow only).

Since according to the definition *Equation 14* the area strain $u$ can be written as, using the coordinates $(s_0, \phi)$ on the undeformed and deformed surfaces and denoting $g_0$ the determinant of the metric of the undeformed surface,

$$u = \frac{\sqrt{g} - \sqrt{g_0}}{\sqrt{g_0}},$$ (40)

we obtain from *Equation 38* the variation:

$$\delta u = (1 + u) \left( \nabla_k \delta X^k + C_k^k \delta X_n \right),$$ (41)

which yields the Lagrangian time derivative (*Equation 15*) in the main text, using $\delta \mathbf{X} = \delta t \mathbf{v}$.

## Axisymmetric surfaces

### Fundamental tensors

On a surface with axial symmetry about the $z$-axis, as defined in the main text, the basis vectors and the outward normal are

$$\boldsymbol{e}_\phi = \begin{pmatrix} -x \sin \phi \\ x \cos \phi \\ 0 \end{pmatrix},$$ (42)

$$\boldsymbol{e}_s = \begin{pmatrix} \cos \phi \cos \psi \\ \sin \phi \cos \psi \\ \sin \psi \end{pmatrix},$$ (43)

$$\boldsymbol{n} = \begin{pmatrix} \cos \phi \sin \psi \\ \sin \phi \sin \psi \\ -\cos \psi \end{pmatrix},$$ (44)

where we have used $\partial_s x = \cos \psi$ and $\partial_s z = \sin \psi$, which can be defined through the requirement that $s$ is an arc length parameter, such that $|\boldsymbol{e}_s|^2 = (\partial_s x)^2 + (\partial_s z)^2 = 1$. The metric and curvature tensors and the surface element are given by

$$g_{ij} = \begin{pmatrix} x^2 & 0 \\ 0 & 1 \end{pmatrix}, \quad C_i^j = \begin{pmatrix} \frac{\sin \psi}{x} & 0 \\ 0 & \partial_s \psi \end{pmatrix}, \quad \mathrm{d}S = x \, \mathrm{d}s \, \mathrm{d}\phi.$$ (45)

In the following, because the metric is diagonal, we will not distinguish between the order of indices for diagonal elements of second-order tensors in mixed coordinates, that is, for a tensor $\mathbf{T}$ we use $T_s^s = T_s^s = T_s^s$ and $T_\phi^\phi = T_\phi^\phi = T_\phi^\phi$. The circumferential and meridional principal curvatures $C_\phi^\phi$ and $C_s^s$, and the mean and Gaussian curvatures $H$ and $K$ are given by:

$$C_\phi^\phi = \frac{\sin \psi}{x}, \quad C_s^s = \partial_s \psi,$$ (46)

$$H = \frac{1}{2} C_k^k, \quad K = \det C_i^j = C_\phi^\phi C_s^s.$$ (47)

Some useful relationships involving the two principal curvatures follow from *Equation 46* and from the definitions *Equations 11 and 12*,

$$\partial_s C_\phi^\phi = \frac{\cos \psi}{x} \left( C_s^s - C_\phi^\phi \right),$$ (48)

$$\partial_s \left( \frac{\cos \psi}{x} \right) = -C_\phi^\phi C_s^s - \left( \frac{\cos \psi}{x} \right)^2 .$$

(49)

The partial area and partial volume are given by

$$a(s) = 2\pi \int_0^s ds' \, x(s'),$$

(50)

$$v(s) = \pi \int_0^s ds' \, x(s')^2 \sin \psi(s') .$$

(51)

The corotational time derivative of the curvature tensor, as defined in **Equation 8**, has trace

$$\frac{DC_k^k}{Dt} = -\partial_s^2 v_n - \frac{\cos \psi}{x} \partial_s v_n - v_n((C_s^s)^2 + (C_\phi^\phi)^2) + v_s \partial_s C_k^k.$$

(52)

## Covariant derivatives

Axial symmetry implies that all functions on the surface should be $\phi$-independent, and we also consider here vector and tensor fields $\mathbf{f}$, $\mathbf{T}$ such that $f^\phi = 0$ and $T^{s\phi} = T^{\phi s} = 0$. The only non-vanishing component of $\partial_i g_{jk}$ is $\partial_s g_{\phi\phi} = 2x \cos \psi$; therefore, the non-zero Christoffel symbols of the first kind are

$$\Gamma_{\phi s\phi} = \Gamma_{\phi\phi s} = -\Gamma_{s\phi\phi} = \frac{1}{2} \partial_s g_{\phi\phi} = x \cos \psi,$$

(53)

and the non-zero Christoffel symbols of the second kind are

$$\Gamma_{\phi\phi}^s = -x \cos \psi, \quad \Gamma_{s\phi}^\phi = \Gamma_{\phi s}^\phi = \frac{\cos \psi}{x}.$$

(54)

The non-zero components of $\nabla_i f^j$ are

$$\nabla_s f^s = \partial_s f^s,$$

(55)

$$\nabla_\phi f^\phi = \frac{\cos \psi}{x} f^s,$$

(56)

resulting in the components of the strain rate tensor defined in **Equation 7**:

$$v_s^s = \partial_s v^s + C_s^s v_n,$$

(57)

$$v_\phi^\phi = \frac{\cos \psi}{x} v^s + C_\phi^\phi v_n,$$

(58)

with trace

$$v_k^k = \partial_s v^s + \frac{\cos \psi}{x} v^s + C_k^k v_n.$$

(59)

The non-zero components of $\nabla_i T^{jk}$ are

$$\nabla_s T^{ss} = \partial_s T^{ss},$$

(60)

$$\nabla_s T^{\phi\phi} = \partial_s T^{\phi\phi} + 2 \frac{\cos \psi}{x} T^{\phi\phi},$$

(61)

$$
\begin{aligned}
\nabla_\phi T^{s\phi} &= -x \cos \psi T^{\phi\phi} + \frac{\cos \psi}{x} T^{ss} \\
&= \frac{\cos \psi}{x} \left( T_s^s - T_\phi^\phi \right) \\
&= \nabla_\phi T^{\phi s},
\end{aligned}
$$

(62)

where $T^{\phi\phi} = g^{\phi\phi}T^\phi_\phi = \frac{1}{x^2}T^\phi_\phi$ was used in the last expression.

## Force and torque balance

Axial symmetry implies that the tangential and the normal force balances, *Equations 1; 2*, and the torque balance *Equation 3* can be rewritten as

$$\partial_s t^s_s = \frac{\cos\psi}{x}(t^\phi_\phi - t^s_s) - C^s_s t^s_n - f^{\text{ext}}_s,$$

(63)

$$\partial_s t^s_n = C^\phi_\phi(t^\phi_\phi - t^s_s) + C^k_k t^s_s - \frac{\cos\psi}{x}t^s_n - f^{\text{ext}}_n - P,$$

(64)

$$\partial_s \bar{m}^s_s = \frac{\cos\psi}{x}(\bar{m}^\phi_\phi - \bar{m}^s_s) + t^s_n.$$

(65)

The geometric singularities appearing in *Equations 63–65* are removed by an appropriate choice of boundary conditions for the tensions and moments at the poles of the surface. The normal torque balance *Equation 4* gives the antisymmetric part of the tension tensor. With the constitutive *Equation 6* for the bending moment tensor, $\epsilon_{ij}t^{ij} = C_{ij}m^{ij} = \zeta_{cn}Q^{ik}\epsilon^j_k C_{ij}$ which vanishes for an axisymmetric surface where $Q^{s\phi} = Q^{\phi s} = 0$. Therefore, here $t^{ij} = t^{ij}_s$ which is given by the constitutive *Equation 5*.

## Direct expression for the transverse tension on an axisymmetric surface

In *Capovilla and Guven, 2002* and *Knoche and Kierfeld, 2011*, it is shown that on axially symmetric surfaces the normal force balance in *Equation 64* can be integrated in a closed form in the presence of a uniform pressure. Here, we generalise this to an arbitrary axially symmetric external force. In analogy to *Capovilla and Guven, 2002*, consider a piece of surface $\mathcal{S}_1$ bounded by the south pole and a circle $\mathcal{C}$ perpendicular to the axis of symmetry, given by $s = s_1$ (*Appendix 1—figure 1*). The bounding circle $\mathcal{C}$ has the line element $dl = xd\phi$ and the unit normal $\nu = \mathbf{e}_s$, tangent to the surface and pointing outward with respect to $\mathcal{S}_1$. The balance of forces acting on $\mathcal{S}_1$ reads

$$\oint_\mathcal{C} dl\nu_i \mathbf{t}^i + \int_{\mathcal{S}_1} d\mathrm{S}\,(\mathbf{f}^{ext} + P\mathbf{n}) = \mathbf{0}.$$

(66)

The contributions to the integral *Equation 66* are

$$\begin{aligned}
\oint_\mathcal{C} dl\nu_i \mathbf{t}^i &= \int_0^{2\pi} d\phi\, x\mathbf{t}^s \\
&= \int_0^{2\pi} d\phi\, x\left(t^{ss}\mathbf{e}_s + t^s_n\mathbf{n}\right) \\
&= 2\pi x\left(t^{ss}\sin\psi - t^s_n\cos\psi\right)\mathbf{e}_z,
\end{aligned}$$

(67)

$$\begin{aligned}
P\int_{\mathcal{S}_1} d\mathrm{S}\,\mathbf{n} &= \left(-2\pi P\int_0^{s_1} ds\, x\cos\psi\right)\mathbf{e}_z \\
&= \left(-2\pi P\int_0^{x(s_1)} dx\, x\right)\mathbf{e}_z \\
&= -\pi Px^2\mathbf{e}_z,
\end{aligned}$$

(68)

$$\begin{aligned}
\int_{\mathcal{S}_1} d\mathrm{S}\,\mathbf{f}^{ext} &= 2\pi\left(\int_0^{s_1} ds\, xf^{ext}_z\right)\mathbf{e}_z \\
&= 2\pi I(s_1)\mathbf{e}_z,
\end{aligned}$$

(69)

where we have introduced the integrated external force

$$I(s) = \int_0^s ds'\, xf^{\text{ext}}_z,$$

(70)

and used the shape *Equation 11* in *Equation 68*. As *Equations 67–69* only contribute to the $z$-component, *Equation 66* can be rewritten as

$$2\pi x \left( t_s^s \sin\psi - t_n^s \cos\psi \right) - \pi P x^2 + 2\pi I = 0. \tag{71}$$

From *Equation 71* one obtains an expression for the transverse tension for $\psi \neq \frac{\pi}{2}$:

$$t_n^s = t_s^s \tan\psi - \frac{1}{2}\frac{x}{\cos\psi}P + \frac{1}{x\cos\psi}I, \tag{72}$$

and it is easy to confirm, using *Equation 63*, that this is indeed a solution of the normal force balance given by *Equation 64*.

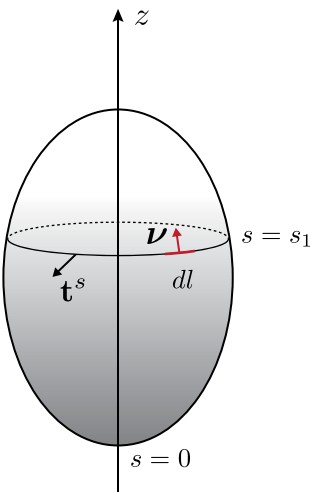

**Appendix 1—figure 1.** Schematic of the surface $\mathcal{S}_1$ used to derive the integral of the normal force balance.

## Behaviour at the poles

The poles of the axisymmetric surface are at $s = 0$ (south pole) and $s = L$ (north pole) and satisfy $x(0) = x(L) = 0$. Besides, we assumed that the shape has a finite curvature, requiring $\psi = 0$ and $\psi = \pi$ at the south and north poles. The asymptotic behaviour of the shape is then:

$$\begin{aligned} x(s) &= x(0) + \cos\psi|_{s=0}s - \frac{1}{2}((\sin\psi)\partial_s\psi)|_{s=0}s^2 + \mathcal{O}(s^3) \\ &= s + \mathcal{O}(s^3), \end{aligned} \tag{73}$$

$$\begin{aligned} x(L-s) &= x(0) + \cos\psi|_{s=L}(L-s) - \frac{1}{2}((\sin\psi)\partial_s\psi)|_{s=L}(L-s)^2 + \mathcal{O}((L-s)^3) \\ &= s - L + \mathcal{O}((L-s)^3). \end{aligned} \tag{74}$$

The limits of geometric singularities of the form $f(s)/x(s)$, for some function $f(s)$ which vanishes at the pole, follow from L'Hôpital's rule

$$\lim_{s\to 0,L}\frac{f(s)}{x(s)} = \lim_{s\to 0,L}\frac{\partial_s f|_{s=0,L}}{\cos\psi(s)|_{s=0,L}} = \pm\partial_s f|_{s=0,L}. \tag{75}$$

where the + sign applies to $s = 0$ and the - sign to $s = L$. For example, applying *Equation 75* to $C_\phi^\phi = \sin\psi/x$ yields that

$$\lim_{s\to 0,L} C_\phi^\phi = C_s^s|_{s=0,L}. \tag{76}$$

Any smooth tangent vector field on the closed axisymmetric surface has to vanish at the poles. For example, since $C_k^k$ is a scalar field, for its derivative we have

$$\partial_s C_k^k|_{s=0,L} = 0. \tag{77}$$

From *Equation 48* we find that at the poles $2\partial_s C_\phi^\phi = \partial_s C_s^s$, which, together with *Equation 77* yields

$$\partial_s C_\phi^\phi|_{s=0,L} = \partial_s C_s^s|_{s=0,L} = 0. \tag{78}$$

This relation, together with *Equation 76*, implies that at the poles, the surface is locally spherical.

## Spherical surface

We give here some of the geometrical quantities defined above for the undeformed initial surface, a sphere with radius $R_0$ and south pole at the origin, $\mathbf{X}(\phi,0) = \mathbf{0}$:

$$L_0 = \pi R_0, \quad 0 \le s \le \pi R_0, \tag{79}$$

$$x(s_0) = R_0 \sin \frac{s_0}{R_0}, \tag{80}$$

$$z(s_0) = R_0 \left( 1 - \cos \frac{s_0}{R_0} \right), \tag{81}$$

$$\psi(s_0) = \frac{s_0}{R_0}, \tag{82}$$

$$C_s^s = C_\phi^\phi = H = \frac{1}{R_0}, \tag{83}$$

$$\frac{\cos \psi}{x} = \frac{1}{R_0 \tan \frac{s_0}{R_0}}, \tag{84}$$

$$a(s_0) = \frac{A_0}{2} \left( 1 - \cos \frac{s_0}{R_0} \right), \tag{85}$$

$$v(s_0) = V_0 \left( 2 + \cos \frac{s_0}{R_0} \right) \sin^4 \frac{s_0}{2R_0}, \tag{86}$$

where the arc length is denoted $s_0$, and $A_0 = 4\pi R_0^2$, $V_0 = (4/3)\pi R_0^3$ are the area and volume of the sphere.

## Appendix 2

### Infinitesimal neck

We discuss here the infinitesimal neck appearing in steady-state shapes subjected to isotropic active bending moments ($\zeta = \zeta_n = \zeta_{cn} = 0$). In that case, the tensors $t^{ij}$ and $\bar{m}^{ij}$ are isotropic and the force and torque balance *Equations 63–65* can be written, in the absence of external force other than the pressure $P$ and for $\psi \neq \frac{\pi}{2}$:

$$\partial_s \left( \frac{t_s^s}{\cos \psi} \right) = \frac{x \partial_s \psi}{2 \cos^2 \psi} P, \tag{87}$$

$$\partial_s \bar{m}_s^s = t_s^s \tan \psi - \frac{1}{2} \frac{x}{\cos \psi} P, \tag{88}$$

where we have used the transverse tension solution (*Equation 72*).

### Scaling analysis

To analyse the behaviour of these equations near an infinitesimal neck, we now perform a scaling analysis following *Fourcade et al., 1994*. We consider a region around a nearly closed neck with minimal radius $a$. At the point of the surface closest to the axis of symmetry, $x = a$ and $\psi = \frac{\pi}{2}$. We then scale the arc length coordinate $s$, the distance of the surface to the axis of symmetry $x$, and the curvature tensor with $a$, and introduce $\bar{s} = s/a$, $\bar{x} = x/a$ and $\bar{C}_k^k = aC_k^k$. The force balance equations then become for $\psi \neq \frac{\pi}{2}$:

$$\partial_{\bar{s}} \left( \frac{t_s^s}{\cos \psi} \right) = \frac{a \bar{x} \partial_{\bar{s}} \psi}{2 \cos^2 \psi} P, \tag{89}$$

$$\partial_{\bar{s}}(2 \kappa \bar{C}_k^k) = -a \partial_{\bar{s}} \zeta_c + a^2 t_s^s \tan \psi - \frac{a^3}{2} \frac{\bar{x}}{\cos \psi} P. \tag{90}$$

For $a \to 0$, the leading order solution has $t_s^s / \cos \psi$ and $\bar{C}_k^k$ both constant. Using the relation

$$\bar{C}_k^k = \partial_{\bar{s}} \psi + \frac{\sin \psi}{\bar{x}} = \frac{1}{\bar{x}} \partial_{\bar{x}} \left( \bar{x}^2 \bar{C}_\phi^\phi \right), \tag{91}$$

with $\bar{C}_\phi^\phi = aC_\phi^\phi$, and the conditions $\bar{C}_\phi^\phi(\bar{x} = 1) = 1$ and that $\sin \psi$ does not diverge for $|\bar{x}| \to \infty$, the curvatures have solution $\bar{C}_k^k = 0$ and

$$C_\phi^\phi = -C_s^s = \frac{a}{x^2}, \cos \psi = \pm \sqrt{1 - \frac{a^2}{x^2}}, \tag{92}$$

where the sign of $\cos \psi$ changes in the regions towards and away from the neck. Therefore, $\cos \psi$ converges to +1 or -1 away from the neck for $x \to \pm\infty$. Since $t_s^s / \cos \psi$ is constant across the infinitesimal neck, $t_s^s$ also changes sign asymptotically away from the neck.

The next order in $a$ of *Equation 90* gives $\partial_{\bar{s}}(2 \kappa C_k^k + \zeta_c) = 0$ which corresponds to $\bar{m}_s^s$ constant across the neck (*Figure 2—figure supplement 1*).

### Analytical solution for the free volume case

In the free volume case, $P = 0$ and the force balance equations admits the solution $t_s^s = u = 0$ and constant $\bar{m}_s^s$ (*Figure 2—figure supplement 1a*). Considering a shape with the active and passive regions forming spheres with radii $R_a$ and $R_p$ separated by an infinitesimal neck, the condition of constant $\bar{m}_s^s$ results in

$$2 \kappa [C_k^k]_a + \delta \zeta_c = 2 \kappa [C_k^k]_p, \tag{93}$$

with $[C_k^k]_a$ and $[C_k^k]_p$ the trace of the curvature tensor in the active and passive regions, and $\delta \zeta_c$ the difference in isotropic active bending moment between the passive and active regions. This results in *Equation 24*, taking into account that $[C_k^k]_a = \pm 2/R_a$ depending on whether the active region is

curved towards the outside or the inside part of the surface. In addition, the condition $u = 0$ results in conservation of area of the active and passive regions compared to the undeformed sphere:

$$4\pi R_a^2 = 2\pi R_0^2 (1 - \cos \frac{l_a}{R_0}),$$
$$4\pi R_p^2 = 2\pi R_0^2 (1 + \cos \frac{l_a}{R_0}),$$

(94)

where we have used *Equation 85*. When $[C_k^k]_a = 2/R_a$ corresponding to the active region towards the outside, and $\delta\zeta_c > 0$, *Equation 24* implies that $R_p < R_a$ which further requires $l_a/L_0 > 1/2$ to satisfy *Equation 94*.

Combining *Equations 93 and 94* gives a condition defining a curve in the parameter space $l_a/L_0$, $\delta\zeta_c R_0/\kappa$:

$$\frac{\delta\zeta_c R_0}{4\kappa} = \sqrt{\frac{2}{1 + \cos\left(\frac{\pi l_a}{L_0}\right)}} \mp \sqrt{\frac{2}{1 - \cos\left(\frac{\pi l_a}{L_0}\right)}},$$

(95)

which agrees well with the line of neck constriction determined numerically (*Figure 2b*).

## Appendix 3

## Nematic order parameter on an axisymmetric surface

### Equilibrium equation

A nematic director on a curved surface is given by a unit tangent vector, for which $\hat{\mathbf{n}} = -\hat{\mathbf{n}}$. In an orthonormal frame $\{\hat{\mathbf{e}}_\phi, \hat{\mathbf{e}}_s\}$, where $\hat{\mathbf{e}}_\phi = \mathbf{e}_\phi/x$ and $\hat{\mathbf{e}}_s = \mathbf{e}_s$, it is characterised by an angle $\alpha \in [0, \pi]$ as

$$\hat{\mathbf{n}} = \cos\alpha\,\hat{\mathbf{e}}_\phi + \sin\alpha\,\hat{\mathbf{e}}_s. \tag{96}$$

The director components in the basis $\{\mathbf{e}_\phi, \mathbf{e}_s\}$ are then

$$\hat{n}^\phi = \frac{\cos\alpha}{x}, \hat{n}^s = \sin\alpha. \tag{97}$$

The traceless and symmetric nematic order parameter $Q^{ij}$ can be constructed from the director $n^i$ and a magnitude $S$ as

$$Q^{ij} = S(\hat{n}^i\hat{n}^j - \frac{1}{2}g^{ij}). \tag{98}$$

Its components read

$$\begin{aligned} Q^{\phi\phi} &= \frac{S}{2}\frac{\cos 2\alpha}{x^2}, \\ Q^{ss} &= -\frac{S}{2}\cos 2\alpha, \\ Q^{s\phi} = Q^{\phi s} &= \frac{S}{2}\frac{\sin 2\alpha}{x}. \end{aligned} \tag{99}$$

We assume here that there is no azimuthal flow on the surface, $v^\phi = 0$. The $\phi$-component of the tangential force balance then reads for constant $\zeta_n \neq 0$, $\zeta_{cn} \neq 0$:

$$\nabla_i t^{i\phi} = \left(\zeta_n + \zeta_{cn}\frac{\sin\psi}{x}\right)\left(\partial_s Q^{s\phi} + 3\frac{\cos\psi}{x}Q^{s\phi}\right) = 0, \tag{100}$$

which requires $Q^{s\phi} = Q^{\phi s} = 0$ excluding divergence of $Q^{s\phi}$ at the poles; therefore, the only possible orientations for the director, compatible with our assumption of vanishing azimuthal flows, are $\alpha = 0, \pi/2$. Therefore, in the axisymmetric setup we consider only one non-zero component

$$q = \frac{S}{2}\cos 2\alpha = Q^\phi_\phi = -Q^s_s, \tag{101}$$

which can take the values $q = \pm S/2$, corresponding to azimuthal or longitudinal orientation of the director $\hat{\mathbf{n}}$, respectively.

The total free energy of the nematic *Equation 9* reads in terms of $q$

$$F = \int \mathrm{d}S f = \int \mathrm{d}S \left(k\left((\partial_s q)^2 + 4\left(\frac{\cos\psi}{x}q\right)^2\right) - \frac{a}{2}q^2 + \frac{a}{4}q^4\right). \tag{102}$$

We minimise this energy with respect to $q$ on a given surface. The resulting Euler–Lagrange *Equation 16* is obtained from

$$0 = \frac{\delta F}{\delta q} = h - \nabla_i \Pi^i, \tag{103}$$

where

$$h = \frac{\partial f}{\partial q}, \quad \Pi^i = \frac{\partial f}{\partial(\partial_i q)}. \tag{104}$$

Here, we have used the definition from the functional derivative, $dF \simeq \int \mathrm{d}S \frac{\delta F}{\delta q} dq$. *Equation 16* can be written as two first-order equations

$$\partial_s q = w, \tag{105}$$

$$\partial_s w = \frac{1}{2l_c^2} q(q^2 - 1) + \frac{\cos\psi}{x}\left(4\frac{\cos\psi}{x}q - w\right). \tag{106}$$

The requirement that *Equation 106* should be regular at the poles of the surface results in the two boundary conditions

$$q(0) = q(L) = 0, \tag{107}$$

and also implies

$$w(0) = w(L) = 0. \tag{108}$$

The limit of *Equation 106* at $s = 0$ is given by

$$
\begin{aligned}
\partial_s w(0) &= \frac{1}{2l_c^2} q(0)(q(0)^2 - 1) + \lim_{s\to 0}\left[\frac{\cos(\psi)}{x}\left(4\frac{\cos(\psi)}{x}q - w\right)\right]\\
&= \frac{1}{2l_c^2} q(0)(q(0)^2 - 1) + \lim_{s\to 0}\left[\frac{1}{s}\left(4\frac{1}{s}\left(q(0) + w(0)s + \frac{1}{2}\partial_s w(0)s^2\right) - (w(0) + \partial_s w(0)s)\right)\right]\\
&= \lim_{s\to 0}\left[\frac{1}{s}\left(3w(0) + \partial_s w(0)s\right)\right]\\
&= \partial_s w(0),
\end{aligned}
\tag{109}
$$

and equivalently at $s = L$. Therefore, *Equation 106* does not provide a limit value for $\partial_s w$ at the poles. When solving *Equations 105 and 106* numerically, we use the analytical limits at the poles

$$\left(\partial_s q, \partial_s w\right)_{s=0} = \left(0, W_0\right), \tag{110}$$

$$\left(\partial_s q, \partial_s w\right)_{s=L} = \left(0, W_L\right), \tag{111}$$

with two free parameters $W_0$ and $W_L$, which are introduced in order to ensure all four boundary conditions *107 and 108*, of which the second two have to be imposed explicitly for numerical reasons.

## Stability of the isotropic state on a sphere

We discuss here the stability of the isotropic state $q = 0$ on a sphere of radius $R_0$ to axisymmetric perturbations; a more general analysis can be found in *Napoli and Vergori, 2012*. We note that for a spherical shape, with $\theta = s/R_0$ and at first order in $q$:

$$\frac{\delta F}{\delta q} \simeq -\frac{2k}{R_0^2}\left[\partial_\theta^2 + \cot\theta\,\partial_\theta - 4\cot^2\theta + \frac{R_0^2}{2l_c^2}\right]q, \tag{112}$$

where we have used *Equation 103* and the geometrical relations for a sphere given in Appendix 1 section 'Spherical surface'. A set of eigenfunctions of the differential operator $\mathcal{L} = \partial_\theta^2 + \cot\theta\,\partial_\theta - 4\cot^2\theta$ is provided by taking derivatives of axisymmetric spherical harmonics, $q_n(\theta) = q_n f_n(\theta)$ with $f_n(\theta) = [\partial_\theta^2 - \cot\theta\,\partial_\theta]P_n(\cos\theta)$ with $P_n$ the Legendre polynomial of degree $n$, for $n \geq 2$. One then finds

$$\frac{\delta F}{\delta q}[q_n] \simeq \frac{2k}{R_0^2}\left[n(n+1) - 4 - \frac{R_0^2}{2l_c^2}\right]q_n f_n(\theta) \tag{113}$$

The isotropic state is stable if $n(n+1) - 4 - \frac{R_0^2}{2l_c^2} \geq 0$ for $n \geq 2$, or for

$$\frac{l_c}{R_o} > \frac{1}{2}. \tag{114}$$

## Appendix 4

### Active nematic tension and bilayered disc: Asymptotic analysis

We discuss here an asymptotic analysis for the flat bilayers disc steady-state shapes found for surfaces subjected to vanishing internal pressure, uniform nematic active tension, and for $\zeta_n < 0$ (*Figure 4b*). We consider here the limit where $\zeta = \zeta_c = \zeta_{cn} = 0$.

We postulate that the limit shape reached as $|\zeta_n| \to \infty$ consists of two parallel flat central discs of radius $R_d$, separated by a distance $2h$, and connected by a narrow curved region (*Appendix 4—figure 1*). One denotes $s_t$ the arclength of the extreme position of the shape where $x = x_t$ is maximal, and $s_c = s - s_t$ the arclength from this point (*Appendix 4—figure 2*). The shape is assumed to be symmetric about a plane going through the equator, which imposes $\partial_s q(s = s_t) = 0$. One looks for an asymptotic shape which satisfies $h \ll R_d$; we show later that this condition requires $\kappa \ll K l_c^2$.

The force and torque balance *Equations 63–65* can then be rewritten

$$\partial_s \left[ \frac{t_s^s}{\cos \psi} \right] = \frac{2\zeta_n q}{x}, \tag{115}$$

$$2\kappa \partial_s \left[ \partial_s \psi + \frac{\sin \psi}{x} \right] = t_s^s \tan \psi. \tag{116}$$

with $t_s^s = 2Ku - \zeta_n q$. The last equation implies that at $\psi = \pi/2$ (i.e. at the point of the surface with the extremal value of $x$), $t_s^s = 0$ and therefore at this point $u = \zeta_n q/(2K)$. However, our definition of deformation implies that $u > -1$. Therefore, as $|\zeta_n| \to \infty$, one must have $q \to 0$ (*Appendix 4—figure 2*), and the equilibrium equation for the nematic order parameter $q$, *Equations 105 and 106* can be linearised.

Introducing a renormalised order parameter $\tilde{q} = -\zeta_n/(2K)q$ and renormalised tension $\tilde{t}_s^s = t_s^s/(2K) = u + \tilde{q}$, the force and torque balance equation and the linearised equilibrium equation for the order parameter read

$$\partial_s \left[ \frac{\tilde{t}_s^s}{\cos \psi} \right] = -\frac{2\tilde{q}}{x}, \tag{117}$$

$$\frac{\kappa}{K} \partial_s \left[ \partial_s \psi + \frac{\sin \psi}{x} \right] = \tilde{t}_s^s \tan \psi, \tag{118}$$

$$\partial_s^2 \tilde{q} + \frac{\cos \psi}{x} \partial_s \tilde{q} - 4 \frac{\cos^2 \psi}{x^2} \tilde{q} = -\frac{1}{2l_c^2} \tilde{q}, \tag{119}$$

to which one can add the boundary conditions $\tilde{q}(0) = \tilde{q}(L) = 0$, $\psi(0) = 0$, $\psi(L) = \pi$ and a condition on $u$ which follows from *Equation 143*:

$$2R_0^2 = \int_0^L \frac{ds\,x}{1+u}. \tag{120}$$

In the asymptotic regime where $h \ll R_d$, we consider separately the flat central discs and the narrow curved connecting region.

### Flat central disc

In the lower, flat part of the deformed shape, one has $\psi = 0$, $x = s$ and *Equation 119* becomes

$$\partial_s^2 \tilde{q} + \frac{1}{s} \partial_s \tilde{q} - \frac{4}{s^2} \tilde{q} = -\frac{1}{2l_c^2} \tilde{q}, \tag{121}$$

with solution, using $\tilde{q}(s = 0) = 0$, $\tilde{q}(x) = C_q J_2 \left( \frac{x}{\sqrt{2}l_c} \right)$, with $C_q$ a constant to determine, and $J_n(x)$ are the Bessel functions of the first kind. The condition $0 = \partial_s q(s = s_t) \simeq \partial_s q(s = R_d)$ then yields the expression for the radius of the disc:

$$R_d = \sqrt{2}l_c \beta_0, \tag{122}$$

with $\beta_0$ the smallest positive solution of $J_2'(\beta_0) = 0$ ($\beta_0 \simeq 3$).

With this solution at hand, solving *Equation 117* gives

$$\tilde{t}_s^s(s) = 2C_q \left( \frac{\sqrt{2}l_c}{s} J_1 \left( \frac{s}{\sqrt{2}l_c} \right) - \frac{1}{\beta_0} J_1(\beta_0) \right), \tag{123}$$

$$u(s) = C_q \left[ 2 \left( \frac{\sqrt{2}l_c}{s} J_1 \left( \frac{s}{\sqrt{2}l_c} \right) - \frac{1}{\beta_0} J_1(\beta_0) \right) - J_2 \left( \frac{s}{\sqrt{2}l_c} \right) \right], \tag{124}$$

using that $0 = \tilde{t}_s^s(s = s_t) \simeq \tilde{t}_s^s(s = R_d)$. The constant $C_q$ can be determined from *Equation 120*:

$$R_0^2 \simeq \int_0^{R_d} \frac{ds\, s}{1 + u(s)}, \tag{125}$$

which can be rewritten:

$$\frac{R_0^2}{l_c^2} \simeq 2 \int_0^{\beta_0} \frac{d\ell\, \ell}{1 + C_q \tilde{u}(\ell)} \tag{126}$$

where one has used $u = C_q \tilde{u}(\frac{s}{\sqrt{2}l_c})$ in the last line, following *Equation 124*. $\tilde{u}(\ell)$ is a decreasing function from $\ell = 0$ to $\ell = \beta_0$ and $\tilde{u}(\beta_0) < 0$. In the limit $l_c/R_0 \to 0$, *Equation 126* is satisfied provided that $C_q \tilde{u}(\beta_0) \to -1$, which sets the constant $C_q$ in that limit. Because $\tilde{u}(\beta_0) < 0$, this implies $C_q > 0$.

In the following, we denote $u_t = u(s_t)$ the deformation reached at the end of the circular plate. Because of the arguments given above, when $l_c/R_0 \to 0$, $u_t \to -1$. In general, $C_q > 0$ implies that $u_t < 0$: we assume this is the case in the following.

## Narrow curved connecting region

In the narrow curved region, at leading order in small $h/R_d$, $\tilde{q} \simeq -u_t = |u_t|$ and $x = x_t \simeq R_d$ are homogeneous, and $|(\sin \psi)/x| \ll |\partial_s \psi|$. The force and torque balance *Equations 117 and 118* now give

$$\partial_s \left( \frac{\tilde{t}_s^s}{\cos \psi} \right) = -\frac{2|u_t|}{x_t}, \tag{127}$$

$$\frac{\kappa}{K} \partial_s^2 \psi = \tilde{t}_s^s \tan \psi, \tag{128}$$

which can be combined to obtain, using $\partial_s \tilde{t}_s^s(s_t) = 0$ because of the symmetry of the shape:

$$\frac{\kappa x_t}{2|u_t|K} \partial_s^2 \psi = -s_c \sin \psi. \tag{129}$$

We look for a solution of this equation $\psi(s_c)$, with the boundary conditions $\psi(s_c = 0) = \frac{\pi}{2}$, $\psi(s_c \to \infty) = \pi$ and, using $\partial_s z = \sin \psi$,

$$\int_0^\infty ds_c \sin \psi(s_c) = h. \tag{130}$$

It is then helpful to introduce the following differential equation:

$$\begin{aligned}
\partial_\ell^2 \tilde{\psi} &= -\ell \sin \tilde{\psi}, \\
\tilde{\psi}(\ell = 0) &= \frac{\pi}{2}, \\
\tilde{\psi}(\ell \to \infty) &= \pi,
\end{aligned} \tag{131}$$

which admits a solution which can be found numerically. The solution of *Equation 129* can then be written

$$\psi(s_c) = \tilde{\psi} \left( s_c \left( \frac{2|u_t|K}{\kappa x_t} \right)^{1/3} \right), \tag{132}$$

and the constraint *Equation 130* gives

$$h \simeq \beta_2 \left[ \frac{\kappa R_d}{2|u_t|K} \right]^{\frac{1}{3}}, \tag{133}$$

where we have introduced $\beta_2 = \int_0^\infty d\ell \sin \tilde{\psi}(\ell)$, with the approximate numerical value $\beta_2 \simeq 1.27$. Together with *Equation 122*, this analysis indicates that the distance $h$ converges to a constant value as $|\zeta_n| \to \infty$ (*Appendix 4—figure 2*). Overall, this analysis predicts the limit value:

$$\frac{2h}{R_d} \simeq \beta_2 \left[ \frac{2\kappa}{|u_t|\beta_0^2 K l_c^2} \right]^{\frac{1}{3}}, \tag{134}$$

such that the condition $h/R_d \ll 1$ implies that the analysis is self-consistent for $\kappa \ll K l_c^2$.

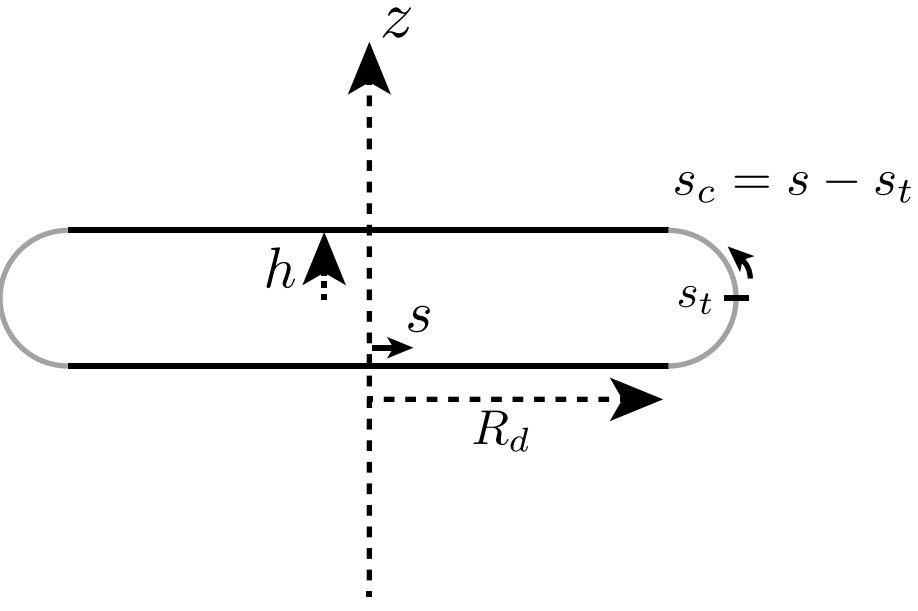

**Appendix 4—figure 1.** Schematic of notations used for the asymptotic analysis of a bilayered disc.

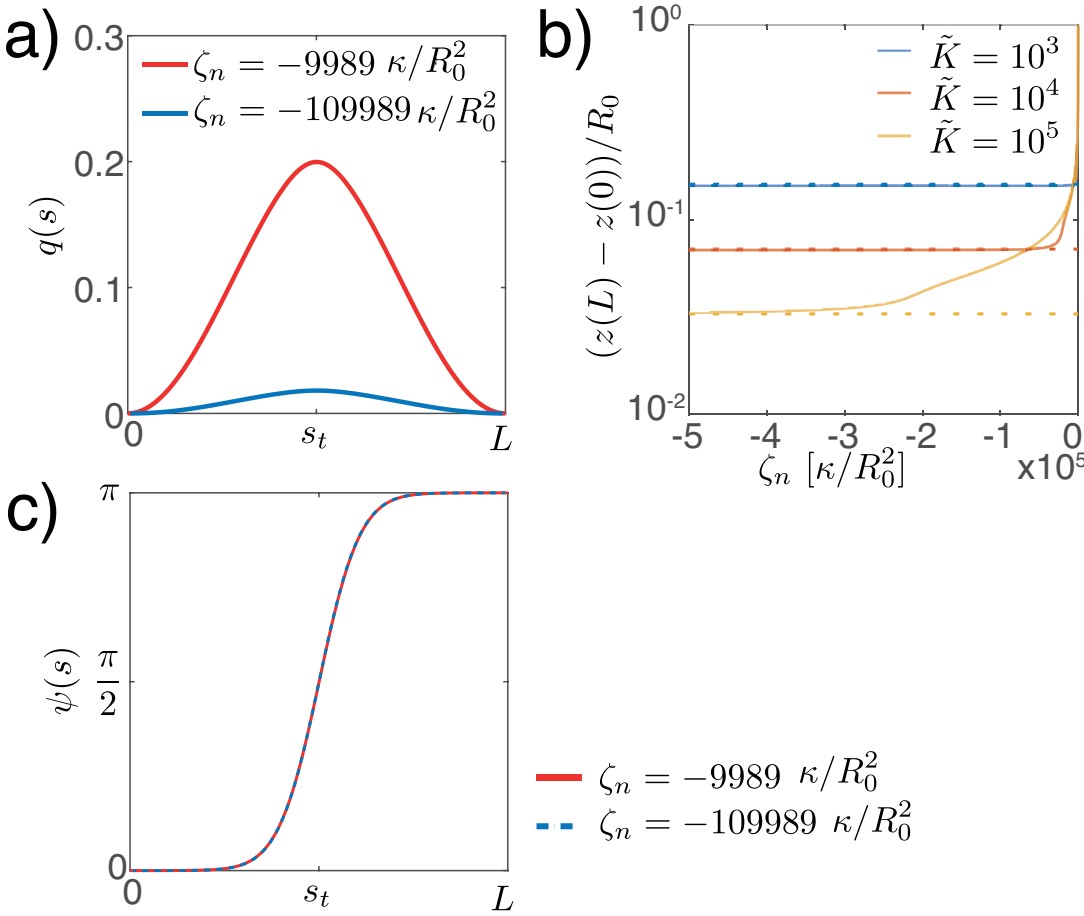

**Appendix 4—figure 2.** Details of shape and nematic profiles for flattened steady-state shapes resulting from a homogeneous nematic tension. (**a**) Profile of nematic order parameter q, which decreases for increasing $|\zeta_n|$. (**b**) Distance between the poles of the steady-state solution for different values of $|\zeta_n|$ and $\tilde{K}$, and corresponding prediction of *Equation 134* (dotted lines). (**c**) Profile of $\psi(s)$ for different values of $\zeta_n$. The profile is invariant with respect to $\zeta_n$, for large values of $|\zeta_n|$.

## Appendix 5

### Numerical method for global force balance

In steady-state and dynamical solving, an overall degree of freedom of solid translation of the surface along its axis of symmetry has to be fixed. In steady-state shape calculations, we impose that the south pole is fixed ($z(s = 0) = 0$). In dynamical simulations, we impose that the centroid of the shape does not move (*Equation 189*). In both cases, these constraints are imposed by introducing a constant dummy force

$$\mathbf{f}^{\text{ext}} = f^c \mathbf{e}_z, \tag{135}$$

which is adjusted to constrain the position of the centre of mass of the shape. The corresponding integrated external force, as defined in *Equation 70*, is

$$I(s) = \int_0^s ds' \, x f^c. \tag{136}$$

Because at low Reynolds number the total force acting on the surface must vanish, the value of $f^c$ should be set to zero by the numerical solver. Indeed, inspection of *Equation 72* shows that the conditions $t_n^s|_{s=0} = t_n^s|_{s=L} = 0$ also imply $I(0) = I(L) = 0$, and since $f^c$ is constant, $f^c = 0$. In practice, $f^c$ deviates slightly from zero due to numerical errors.

# Appendix 6

## Numerical methods to determine steady-state shapes

### Stationary shape equations and boundary conditions

The system of stationary shape equations comprises, using the force and torque balance *Equations 63–65*, the equilibrium *Equations 105 and 106* for the nematic order parameter, the definition of the strain *Equation 14* and geometrical relations introduced in Appendix 1:

$$\partial_s t_s^s = 2\frac{\cos\psi}{x}\zeta_n q - C_s^s t_n^s - f^c \sin\psi \tag{137}$$

$$\partial_s t_n^s = 2C_\phi^\phi \zeta_n q + C_s^s t_s^s - \frac{P}{2} - \frac{I}{x^2} + f^c \cos\psi \tag{138}$$

$$\partial_s C_k^k = \frac{1}{2\kappa}\left(t_n^s - \partial_s \zeta_c + (\partial_s \zeta_{cn})q + \zeta_{cn}\left(\partial_s q + 2\frac{\cos\psi}{x}q\right)\right) \tag{139}$$

$$\partial_s \psi = C_s^s, \tag{140}$$

$$\partial_s x = \cos\psi, \tag{141}$$

$$\partial_s z = \sin\psi, \tag{142}$$

$$\partial_s s_0 = \frac{x}{x_0(u+1)}, \tag{143}$$

$$\partial_s q = w, \tag{144}$$

$$\partial_s w = \frac{1}{2l_c^2}q(q^2 - 1) + \frac{\cos\psi}{x}\left(4\frac{\cos\psi}{x}q - w\right), \tag{145}$$

$$\partial_s I = xf^c, \tag{146}$$

where $C_\phi^\phi = \frac{\sin\psi}{x}$, $C_s^s = C_k^k - C_\phi^\phi$, $x_0 = x_0(s_0(s)) = R_0 \sin(s_0(s)/R_0)$, $u = (t_s^s + \zeta_n q - \zeta)/2K$. Here, *Equation 138* has been obtained by using *Equation 72* to rewrite the normal force balance (*Equation 64*) as

$$\partial_s t_n^s = C_\phi^\phi(t_\phi^\phi - t_s^s) + C_s^s t_s^s - \frac{P}{2} - \frac{I}{x^2} + f^c \cos\psi, \tag{147}$$

with the integral of the force $I$ defined in *Equation 136*. The term $I/x^2$ in *Equation 147* is regular at the poles by construction since it has the Taylor expansions $I(s) = \frac{f^c}{2}s^2 + \mathcal{O}(s^3)$ at the south pole and $I(L-s) = -\frac{f^c}{2}(L-s)^2 + \mathcal{O}((L-s)^3)$ at the north pole. Therefore, the limits are $\lim_{s\to 0} I/x^2 = \frac{f^c}{2}$ and $\lim_{s\to L} I/x^2 = -\frac{f^c}{2}$.

The geometric boundary conditions for the shape are $\psi(0) = x(0) = x(L) = z(0) = 0$, $\psi(L) = \pi$. Further, we require $t_n^s(0) = 0$, $I(0) = I(L) = 0$, and $s_0(0) = 0$. The equations for the nematic require $q(0) = q(L) = 0$ and $w(0) = w(L) = 0$, as discussed in Appendix 3. If $l_a = L_0$, the unknown length $L$ is determined from the condition $s_0(L) = L_0$. If $0 < l_a < L_0$, the domain of integration is split into $s \in [0, L_1]$ and $s \in [L_1, L]$. The additional condition $s_0(L_1) = l_a$ sets $L_1$. All unknown functions are matched at the internal boundary $s = L_1$, except for the curvature and strain. These may acquire a jump, $C_s^s(l_a^+) - C_s^s(l_a^-) = (\delta\zeta_c - \delta\zeta_{cn}q(l_a))/2\kappa$ and $u(l_a^+) - u(l_a^-) = (\delta\zeta - \delta\zeta_n q(l_a))/2K$, to ensure continuity in $\bar{m}_s^s$ and $t_s^s$.

If applicable, it is convenient to formulate the volume constraint $V = V_0$ as a boundary value problem for the partial volume $v(s)$ defined in *Equation 51*,

$$\partial_s v = \pi x^2 \sin\psi, \quad v(0) = 0, \quad v(L) = V_0. \tag{148}$$

Due to the geometric singularities appearing in several of the equations at the poles of the surface $s = 0$ and $s = L$, we derive the limits of these equations there. Denoting the solution vector

by $\mathbf{x}(s) = \left(t_s^s, t_n^s, C_k^k, \psi, x, z, s_0, q, w, I, v\right)$, the limiting expressions of *Equations 137–148* at the south and north poles, respectively, are

$$\lim_{s \to 0} \partial_s \mathbf{x} = \left(0, \frac{1}{2}\left(C_k^k(0)t_s^s(0) - P + f^c\right), 0, \frac{C_k^k(0)}{2}, 1, 0, \frac{1}{\sqrt{1+u(0)}}, 0, W_0, 0, 0\right), \tag{149}$$

$$\lim_{s \to L} \partial_s \mathbf{x} = \left(0, \frac{1}{2}\left(C_k^k(L)t_s^s(L) - P - f^c\right), 0, \frac{C_k^k(L)}{2}, -1, 0, \frac{1}{\sqrt{1+u(L)}}, 0, W_L, 0, 0\right). \tag{150}$$

In summary, when considering the full interval and conserved volume, the boundary conditions are

$$\text{at } s = 0 : t_n^s = 0, x = z = \psi = 0, v = 0, s_0 = 0, q = 0, w = 0, I = 0, \tag{151}$$

$$\text{at } s = L : x = 0, \psi = \pi, v = V_0, s_0 = L_0, q = 0, w = 0, I = 0, \tag{152}$$

and the free parameters are $L, P, f^c, W_0$, and $W_L$. Otherwise, if the volume is free to change, then $P = 0$ and the condition $v(L) = V_0$ is removed.

In the dimensionless equations, arc lengths are transformed to the unit interval by $\tilde{s} = s/L$ if $l_a/L_0 = 1$, and to two unit intervals in the case of step-profiles. With $L_1$ and $L_2 = L - L_1$, the dimensionless variables are $\tilde{s} \in [0, 1]$ with $s = \tilde{s}L_1$ in the first interval and $\tilde{s} \in [1, 2]$ with $s = L_1 + L_2(\tilde{s} - 1)$ in the second. The boundary value problem given by *Equations 137–148* and conditions *Equations 151 and 152* in their dimensionless form is solved with the bvp4c solver of MATLAB. The relative and absolute tolerances used in simulations are $tol_{rel} = 10^{-4}$ and $tol_{ans} = 10^{-6}$, leading to typical adaptive grid sizes of $ngrid = 100 - 500$, depending on the shape of the surface. For efficiency, we provide the solver with the analytical Jacobians for the main equations, for the limits at the poles, and for the boundary conditions with respect to the unknowns and the free parameters.

## Construction of solution branches

A solution of the mechanical equilibrium equations can be represented by a vector $\mathbf{p} \in \mathbb{R}^N$, where $\mathbb{R}^N$ is the vector space spanned by all $N$ (or a subset of) parameters of the model, for example, $P$ and $L$, the boundary values of the curvature, tension, etc., and the control parameter. A gradual change of the control parameter corresponds to moving along a solution branch in this parameter space. For small increments in the control parameter, the new solution can be obtained numerically using the previous solution as the initial guess for the solver. However, in many cases the solution branch has a fold (e.g. see *Figure 2b*) and becomes multivalued as a function of the control parameter so that the above method cannot be used.

To continue a solution branch after a fold, we implement a parametric curve approach instead. We denote by $\mathbf{p}^{(i)}$ the current state and by $\mathbf{p}^{(i-1)}$ the previous state of the system and approximate by $\hat{\mathbf{t}}^{(i)} = (\mathbf{p}^{(i)} - \mathbf{p}^{(i-1)})/|\mathbf{p}^{(i)} - \mathbf{p}^{(i-1)}|$ the tangent vector at the current state. To find a new solution $\mathbf{p}^{(i+1)}$ on the curve, a step of length $l_s$ in the direction of the tangent is taken and the new solution is constrained to lie in a (hyper-)plane perpendicular to the tangent, that is,

$$\left(\left(\mathbf{p}^{(i+1)} - \mathbf{p}^{(i)}\right) - l_s\hat{\mathbf{t}}^{(i)}\right) \cdot \hat{\mathbf{t}}^{(i)} = 0, \tag{153}$$

which allows us to eliminate one (e.g. the first) component of $\mathbf{p}^{(i+1)}$ via

$$p_1^{(i+1)} = p_1^{(i)} + \frac{l_s - \left(\mathbf{p}_{2,\cdots,N}^{(i+1)} - \mathbf{p}_{2,\cdots,N}^{(i)}\right) \cdot \hat{\mathbf{t}}_{2,\cdots,N}^{(i)}}{\hat{t}_1^{(i)}}, \tag{154}$$

provided that $\hat{t}_1^{(i)} \neq 0$ and where $\mathbf{p}_{2,\cdots,N}^{(i+1)}$ denotes the vector $\mathbf{p}^{(i+1)}$ without the first component, etc. The remaining $N - 1$ parameters are determined by the solver such that the new point $\mathbf{p}^{(i+1)}$ is a continuation of the solution branch, which can be achieved for small enough $l_s$. It suffices to include only a subset of the free parameters in this construction, for example, $\mathbf{p} = (\delta\zeta_c, P, L_1)$ for a step-like profile of active moments with conserved volume. Since the elimination (*Equation 154*) introduces new dependencies of the differential equations and boundary conditions on the free parameters, the analytical Jacobians for the solver are adjusted accordingly.

# Appendix 7

## Numerical method for the dynamics of active shells

### Force and torque balance equations and boundary conditions

The force and torque balance *Equations 63–65*, together with the constitutive *Equations 5–8* and with $\mathbf{f}^{ext} = f^c \mathbf{e}_z$, can be written as

$$
\begin{aligned}
\partial_s^2 v^s = \quad & -\frac{\cos\psi}{x}\left(\partial_s v^s - \frac{\cos\psi}{x}v^s\right) \\
& -v_n\partial_s C_k^k - \frac{\eta-\eta_b}{\eta+\eta_b}C_s^s C_\phi^\phi v^s \\
& -\frac{1}{\eta+\eta_b}(\eta_b C_k^k + \eta(C_s^s - C_\phi^\phi))\partial_s v_n \\
& +\frac{1}{\eta+\eta_b}\left(-2K\partial_s u - \partial_s\zeta + (\partial_s\zeta_{\mathrm{n}})q + \zeta_{\mathrm{n}}(\partial_s q)\right. \\
& \left. +2\zeta_{\mathrm{n}}q\frac{\cos\psi}{x} - C_s^s t_n^s - f^c\sin\psi\right),
\end{aligned}
\tag{155}
$$

$$
\partial_s t_n^s = C_\phi^\phi\left(t_\phi^\phi - t_s^s\right) + C_k^k t_s^s - \frac{\cos\psi}{x}t_n^s - P + f^c\cos\psi,
\tag{156}
$$

$$
\partial_s \bar{m}_s^s = 2\zeta_{cn}q\frac{\cos\psi}{x} + t_n^s,
\tag{157}
$$

$$
\begin{aligned}
\partial_s^2 v_n = \quad & -\frac{\cos\psi}{x}\partial_s v_n - \left(\left(C_\phi^\phi\right)^2 + \left(C_s^s\right)^2\right)v_n \\
& +v^s\partial_s C_k^k - \frac{1}{\eta_{cb}}\left(\bar{m}_s^s - 2\kappa C_k^k - \zeta_c + \zeta_{cn}q\right),
\end{aligned}
\tag{158}
$$

where in *Equation 156* one has to replace

$$
\begin{aligned}
t_s^s = \quad & 2Ku + \zeta - \zeta_{\mathrm{n}}q \\
& +\left(\eta + \eta_b\right)\partial_s v^s + \left(\eta_b - \eta\right)\frac{\cos\psi}{x}v^s \\
& +\left(\eta_b C_k^k + \eta\left(C_s^s - C_\phi^\phi\right)\right)v_n,
\end{aligned}
\tag{159}
$$

$$
t_\phi^\phi - t_s^s = 2\left(\zeta_{\mathrm{n}}q + \eta\left(\frac{\cos\psi}{x}v^s - \partial_s v^s + \left(C_\phi^\phi - C_s^s\right)v_n\right)\right).
\tag{160}
$$

The solution vector

$$
\mathbf{x}(s) = \left(v^s, \partial_s v^s, t_n^s, \bar{m}_s^s, v_n, \partial_s v_n\right)
\tag{161}
$$

is determined from

$$
\partial_s \mathbf{x} = \mathcal{L}[\mathbf{x}],
\tag{162}
$$

where the linear operator $\mathcal{L}$ is constructed from *Equations 155–158* together with two trivial relations relating $\partial_s v^s$ to $v^s$ and $\partial_s v^n$ to $v^n$. The system of ode's (*Equation 162*) is solved on the full interval $[0, L]$ with the boundary conditions

$$
\text{at } s = 0: \quad v^s = 0, t_n^s = 0, \partial_s v_n = 0,
\tag{163}
$$

$$
\text{at } s = L: \quad v^s = 0, t_n^s = 0, \partial_s v_n = 0.
\tag{164}
$$

They follow from the requirement that any tangent vector field on the closed axisymmetric surface has to vanish at the poles. Equivalently, these are the conditions required to remove the geometric singularities that appear at the poles in *Equations 155–158*. We can then derive the well-defined limits

$$\lim_{s \to 0} \partial_s \mathbf{x} = \left( \partial_s v^s(0), 0, \frac{1}{2} \left( C_k^k(0) t_s^s(0) - P + f^c \right), 0, 0, -\frac{1}{4} C_k^k(0) v_n(0) - \frac{\bar{m}_s^s(0) - 2\kappa C_k^k(0) - \zeta_c(0)}{2\eta_{cb}} \right),$$

(165)

$$\lim_{s \to L} \partial_s \mathbf{x} = \left( \partial_s v^s(L), 0, \frac{1}{2} \left( C_k^k(L) t_s^s(L) - P + f^c \right), 0, 0, -\frac{1}{4} C_k^k(L) v_n(L) - \frac{\bar{m}_s^s(L) - 2\kappa C_k^k(L) - \zeta_c(L)}{2\eta_{cb}} \right),$$

(166)

where

$$t_s^s(0) = 2Ku(0) + \zeta(0) + (\eta + \eta_b)\partial_s v^s(0) + \eta_b C_k^k(0) v_n(0),$$

(167)

$$t_s^s(L) = 2Ku(L) + \zeta(L) + (\eta + \eta_b)\partial_s v^s(L) + \eta_b C_k^k(L) v_n(L).$$

(168)

In *Equations 165 and 166*, we have used that $\zeta$ and $u$, defined as continuous functions on the closed surface, satisfy

$$\partial_s \zeta(0) = \partial_s \zeta(L) = 0, \ \partial_s u(0) = \partial_s u(L) = 0.$$

(169)

Prior to solving system *Equation 162*, at every time step the nematic profiles $q$ and $\partial_s q = w$ are determined on the shape $\mathbf{X}(\phi, s, t)$ as solutions of the Euler–Lagrange *Equation 16*,

$$\partial_s q = w,$$

(170)

$$\partial_s w = \frac{1}{2l_c^2} q(q^2 - 1) \frac{\cos \psi}{x} \left( 4 \frac{\cos \psi}{x} q - w \right),$$

(171)

with the boundary conditions $q(0) = q(L) = w(0) = w(L) = 0$, as discussed in Appendix 3.

## Constraints
### Volume
The volume of a closed surface reads

$$V = \frac{1}{3} \oint dS \, \mathbf{X} \cdot \mathbf{n}.$$

(172)

According to *Equations 34, 36, and 38*, for a Lagrangian surface update with $\partial_t \mathbf{X} = v^i \mathbf{e}_i + v_n \mathbf{n}$ we have

$$\partial_t \sqrt{g} = \sqrt{g} \left( \nabla_i v^i + C_i^i v_n \right),$$

(173)

$$\partial_t \mathbf{n} = \left( -\partial_i v_n + C_{ij} v^j \right) \mathbf{e}^i.$$

(174)

This allows to calculate from *Equation 172* the rate of change in volume

$$\partial_t V = \frac{1}{3} \oint ds^1 ds^2 \partial_t \left( \sqrt{g} \mathbf{X} \cdot \mathbf{n} \right)$$

(175)

$$= \frac{1}{3} \oint dS \left( v_n + \left( \nabla_i v^i + C_i^i v_n \right) \mathbf{X} \cdot \mathbf{n} + \left( -\partial_i v_n + C_{ij} v^j \right) \left( \mathbf{X} \cdot \mathbf{e}^i \right) \right)$$

(176)

$$= \frac{1}{3} \oint dS \left( v_n + C_i^i v_n \mathbf{X} \cdot \mathbf{n} + C_{ij} v^j \left( \mathbf{X} \cdot \mathbf{e}^i \right) - v^k C_k^i \left( \mathbf{X} \cdot \mathbf{e}_i \right) + v_n \left( 2 - C_i^i \mathbf{X} \cdot \mathbf{n} \right) \right)$$

(177)

$$= \oint dS \, v_n,$$

(178)

where we have used two integrations by part and the relations $\partial_k (\mathbf{X} \cdot \mathbf{n}) = \mathbf{X} \cdot C_k^i \mathbf{e}_i$, and $\frac{\partial_i (\sqrt{g} \mathbf{X} \cdot \mathbf{e}^i)}{\sqrt{g}} = 2 - C_i^i \mathbf{X} \cdot \mathbf{n}$.

If the volume is fixed then for all $t$ the dynamics has to satisfy

$$\partial_t V = 0. \tag{179}$$

On the axisymmetric surface, this results in the integral constraint $\partial_t V = 2\pi \int ds\, x v_n$. We define a partial rate of volume change $r_v(s) = 2\pi \int_0^s ds'\, x v_n$, such that the constraint *Equation 179* can be written as

$$\partial_s r_v = 2\pi x v_n, \quad r_v(0) = r_v(L) = 0. \tag{180}$$

This is solved simultaneously with the boundary value problem (*Equations 162–168*), where the pressure $P$ is a free parameter which is required to satisfy both conditions in *Equation 180*.

On the other hand, if the volume is free to change then pressure is no longer a free parameter, but instead couples the normal force balance (*Equation 156*) to the rate of change of volume via *Equation 21*, which can be written:

$$P = -\eta_V r_v(L). \tag{181}$$

In this case $P \to 0$ as the dynamics simulation approaches a steady state, in agreement with the direct steady-state calculations in which $P = 0$.

## Rigid-body translation

We note that in the absence of external force, for any flow profile $\mathbf{v}(s)$ solution of the force and torque balance *Equations 155–158*, the flow

$$\mathbf{v}' = \mathbf{v} + a\mathbf{e}_z \tag{182}$$

$$= \mathbf{v} + a \sin \psi(s) \mathbf{e}_s - a \cos \psi \mathbf{n} \tag{183}$$

with an arbitrary constant $a$, is again a solution. The addition of the uniform flow field $a\mathbf{e}_z$ corresponds to a rigid-body translation in the $z$-direction which does not affect force balance and therefore makes the task of numerically determining $\mathbf{v}$ ill-posed.

To remove this degree of freedom we introduce in each time step a constraint on the translation speed of the centroid. The centroid is equivalent to the centre of mass for a surface with uniform density and is defined as

$$\mathbf{X}_c = \frac{\oint dS\, \mathbf{X}(\phi, s)}{\oint dS}. \tag{184}$$

For a Lagrangian surface update one obtains

$$\partial_t \left( \oint dS\, \mathbf{X} \right) = \oint dS\, \left( \mathbf{X} \frac{\partial_t \sqrt{g}}{\sqrt{g}} + \partial_t \mathbf{X} \right)$$
$$= \oint dS\, \left( v_n C_i^i \mathbf{X} + v_n \mathbf{n} \right), \tag{185}$$

$$\partial_t \left( \oint dS \right) = \oint dS\, \frac{\partial_t \sqrt{g}}{\sqrt{g}} \tag{186}$$

$$= \oint dS\, v_n C_i^i, \tag{187}$$

where we have used *Equation 174* to obtain $\partial_t(\sqrt{g})/\sqrt{g}$, and the divergence theorem on curved surfaces (*Equation 33*). It follows that the velocity of the centroid is given by

$$\partial_t \mathbf{X}_c = \frac{1}{\oint dS} \left( \partial_t \left( \oint dS\, \mathbf{X} \right) - \mathbf{X}_c \partial_t \left( \oint dS \right) \right)$$
$$= \frac{1}{\oint dS} \left( \oint dS\, v_n \left( C_i^i (\mathbf{X} - \mathbf{X}_c) + \mathbf{n} \right) \right) \tag{188}$$

and we want to fix

$$\partial_t \mathbf{X}_c = \mathbf{0} \tag{189}$$

for all $t$. We note that tangential velocity components do not contribute to the centroid displacement.

On the axisymmetric surface $\mathbf{X}_c = z_c \mathbf{e}_z$ and the constraint *Equation 189* becomes

$$\partial_t z_c = \frac{1}{\int ds\, x} \left( \int ds\, x v_n \left( C_i^i (z - z_c) - \cos \psi \right) \right) = 0. \tag{190}$$

Analogously to the volume constraint, we introduce a partial centroid velocity $r_c(s) = \int_0^s ds'\, x v_n \left( C_i^i (z - z_c) - \cos(\psi) \right)$ and add the constraint *Equation 190* to the boundary value problem (*Equations 162–168*) in the form of

$$\partial_s r_c = x v_n \left( C_i^i (z - z_c) - \cos \psi \right), \quad r_c(0) = r_c(L) = 0. \tag{191}$$

Note that due to the free parameter $f^c$ the number of boundary conditions in the full problem, comprising *Equations 162–168 and 191*, is consistent.

## Time update of the surface

The surface update in each time step $t \to t + \delta t$ consists of two sub-steps: first, the material points are advected using a Lagrangian coordinate $s$ as given in *Equation 23*, then the surface $\mathbf{X}'(s, t + \delta t)$ is reparametrised in a new arc length coordinate $s'$, for which $g_{s's'} = 1$.

From *Equation 22* we find the time updates for the shape descriptors:

$$
\begin{aligned}
x(s, t + \delta t) &= x(s, t) + \delta t \left( v_n \sin \psi + v^s \cos \psi \right), \\
z(s, t + \delta t) &= z(s, t) + \delta t \left( v^s \sin \psi - v_n \cos \psi \right), \\
\psi(s, t + \delta t) &= \psi(s, t) + \delta t \left( -\partial_s v_n + v^s C_s^s \right),
\end{aligned}
\tag{192}
$$

where the last equation can be shown by taking the time derivative of the normal vector $\mathbf{n}$ and using *Equation 36*. It is convenient to derive the time update for $C_k^k$ and its derivative from the constitutive *Equation 6* and the torque balance (*Equation 65*),

$$C_k^k(s, t + \delta t) = C_k^k(s, t) + \delta t \frac{1}{\eta_{cb}} \left( \bar{m}_s^s - 2\kappa C_k^k - \zeta_c + \zeta_{cn} q \right), \tag{193}$$

$$\partial_s C_k^k(s, t + \delta t) = \partial_s C_k^k(s, t) + \delta t \frac{1}{\eta_{cb}} \left( t_n^s + 2\zeta_{cn} q \frac{\cos \psi}{x} + \partial_s(-2\kappa C_k^k - \zeta_c + \zeta_{cn} q) \right) \tag{194}$$

because the trace of the corotational derivative (*Equation 8*) is the Lagrangian time evolution of $C_k^k$, as can be seen by using *Equation 39* with $\delta \mathbf{X} = \delta t \mathbf{v}$. From the variation (*Equation 39*) the circumferential curvature is updated as

$$C_\phi^\phi(s, t + \delta t) = C_\phi^\phi(s, t) + \delta t \left( -\partial_s v_n \frac{\cos \psi}{x} - v_n \left( C_\phi^\phi \right)^2 + v^s \partial_s C_\phi^\phi \right) \tag{195}$$

and its derivative is obtained from relation *Equation 48*. In this way only functions which are part of the solution vector (*Equation 161*) are required for the time updates and we avoid taking numerical gradients of the shape or the velocity fields. Finally, according to *Equation 15* we update the area strain as

$$u(s, t + \delta t) = u(s, t) + \delta t (1 + u) v_k^k, \tag{196}$$

with the trace of the strain rate tensor given in *Equation 59*.

The displaced surface is reparametrised in a new arc length coordinate $s'$,

$$\mathbf{X}'(s) = \mathbf{X}'(s'), \tag{197}$$

such that $g_{s's'} = 1$, or equivalently $|\partial_{s'} \mathbf{X}'(s')| = 1$. From

$$\begin{aligned}
\partial_{s'}\mathbf{X}' &= \frac{\partial s}{\partial s'}\partial_s \mathbf{X}' \\
&= \frac{\partial s}{\partial s'}\left[\partial_s \mathbf{X} + \delta t \partial_s \left(v^s \mathbf{e}_s + v_n \mathbf{n}\right)\right] \\
&= \frac{\partial s}{\partial s'}\left[\mathbf{e}_s\left(1 + \delta t\left(v_n C_s^s + \partial_s v^s\right)\right) + \mathbf{n}\delta t(\partial_s v_n - v^s C_s^s)\right]
\end{aligned} \tag{198}$$

we obtain

$$\left(\frac{\partial s'}{\partial s}\right)^2 = 1 + 2\delta t\left(v_n C_s^s + \partial_s v^s\right), \tag{199}$$

so the relationship between the two arc length parameters is given by the differential equation

$$\partial_s s' = 1 + \delta t\left(v_n C_s^s + \partial_s v^s\right). \tag{200}$$

We rewrite this equation in terms of a rate of change of arc length $r_s = \partial_t \left(s' - s\right)$,

$$\partial_s r_s = v_n C_s^s + \partial_s v^s, \quad r_s(0) = 0, \tag{201}$$

and it is added to the linear system (*Equations 162–164*). The new arc length is obtained as $s' = s + r_s \delta t$ and the perimeter length is updated as

$$L(t + \delta t) = L(t) + r_s(L(t))\delta t. \tag{202}$$

For the active profiles defined via sigmoid functions, as given in the main text in *Equation 18*, the parameters chosen for the simulations are $\mu_t = 0.01\tau_a$, $\sigma_t = 0.002\tau_a$, $\mu_s = l_a$, and $\sigma_s = 0.005L_0$. In the time step $t \to t + \delta t$ the bending moment profile, as defined at time $t = 0$ via *Equation 19*, is updated as

$$\zeta_c(s', t + \delta t) = (1 - f(t + \delta t, \mu_t, \sigma_t))(\zeta_c^0 + \delta\zeta_c f(s_0(s'), l_a, \sigma_s)), \tag{203}$$

and an analogous relation holds for $\zeta$, $\zeta_n$, and $\zeta_{cn}$. For the spatial dependence, we keep track of the arc length on the initial sphere $s_0(s')$ as a function of the arc length on the deformed surface after reparametrisation.

Finally, all surface quantities are saved as spline interpolants on the new arc length $s'$. For example,

$$C_k^k(s', t + \delta t) = C_k^k(s(s'), t + \delta t), \tag{204}$$

$$\partial_{s'} C_k^k(s', t + \delta t) = \frac{\partial s}{\partial s'}\partial_s C_k^k(s(s'), t + \delta t), \tag{205}$$

where the Jacobian prefactor for the derivative is given by *Equation 200*.

The size of the adaptive time step $\delta t$ is determined using a standard step doubling method (*Press et al., 2007*), where the shape after a full time step, denoted by $\mathbf{X}^{(1)}(t + \delta t)$, is compared to that after two half steps, denoted by $\mathbf{X}^{(2)}(t + \delta t)$. The relative error is calculated from the shape components and the curvature derivative as

$$\varepsilon_t = \max\left\{\left|\frac{x^{(1)} - x^{(2)}}{x^{(1)}}\right|, \left|\frac{z^{(1)} - z^{(2)}}{z^{(1)}}\right|, \left|\frac{\partial_s(C^{(1)})_k^k - \partial_s(C^{(2)})_k^k}{\partial_s(C^{(1)})_k^k}\right|\right\} \tag{206}$$

on a uniform grid which is defined on $\mathbf{X}(t)$ and projected onto $\mathbf{X}^{(1)}$ and $\mathbf{X}^{(2)}$, whereby the poles and values of $z$ and $\partial_s C_k^k$, which are too close to zero ($< 10^{-3}$), are excluded.

## Numerical convergence to steady state

In order to validate our simulation method for the dynamics of active surfaces, we analyse the numerical convergence of the dynamics simulations to the steady states obtained from direct calculation (see Appendix 6) for the different active effects. A dynamics simulation is regarded as relaxed to steady state when $\max\{|\tilde{v}_n|\} < 10^{-4}$, which defines $t_{relax}$. As an example we show in *Appendix 7—figure 1* the convergence results for the active bending profile

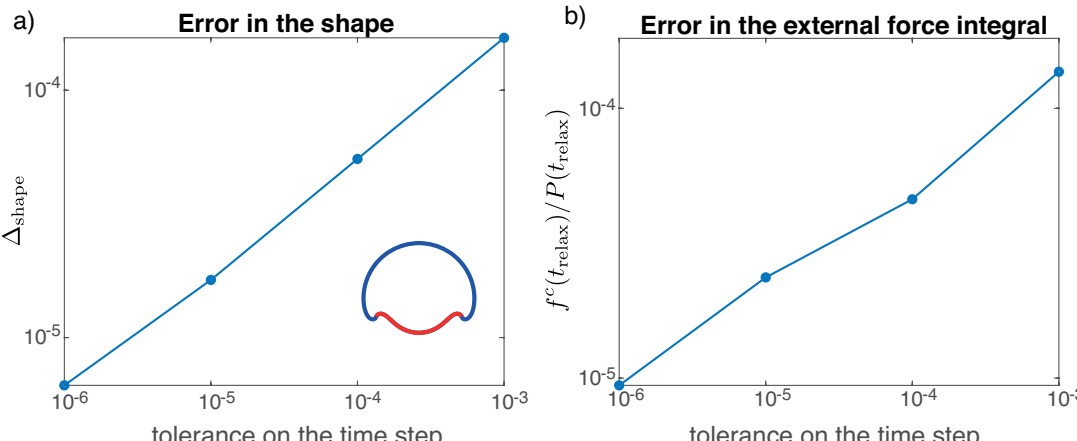

**Appendix 7—figure 1.** Convergence analysis of a dynamics simulation to a steady state obtained from direct calculation, for the example shape shown in the inset of (**a**). For different $tol_t$ (time step) the error in the shape in (**a**) and error in the external force integral in (**b**) are shown.

$$\zeta_c(s_0) = \delta\zeta_c f(s_0, l_a, \sigma_s) \tag{207}$$

with $\delta\zeta_c = 80.73\kappa/R_0$, $l_a = 0.3L_0$, and conserved volume. This profile results in the folded shape shown in the inset of *Appendix 7—figure 1a*. The error in the shape (in units of $R_0$) is calculated as

$$\Delta_{shape} = \frac{1}{N}\sum_i \sqrt{\left(\tilde{x}_i^{dyn} - \tilde{x}_i^{steady}\right)^2 + \left(\tilde{z}_i^{dyn} - \tilde{z}_i^{steady}\right)^2} \tag{208}$$

on a dimensionless, uniform grid $\tilde{s}_i \in [0,1]$, $i = 1, \cdots, N$, with $N = 1000$, which is obtained through division by $L$ or $L(t_{relax})$, respectively, from the corresponding simulation. As discussed in Appendix 5, the deviation of the parameter $f^c$ (see *Equation 135*) from zero characterises the numerical error in the global force balance. As a relative error we plot the value of the parameter $f^c(t_{relax})$ normalised by the pressure $P(t_{relax})$. We find good convergence of the dynamics to the steady-state result with decreasing tolerance $tol_t$ on the time step. Based on these results, we take $tol_t = 10^{-4}$ for the dynamics simulations shown in the main text. The relative and absolute tolerances for the bvp solver are chosen to be the same as for the direct steady-state calculations: $tol_{rel} = 10^{-4}$, $tol_{abs} = 10^{-6}$.

## Appendix 8

### Isotropic active tension

Consider first a shell with vanishing internal hydrostatic pressure, $P = 0$, and a step-like tension profile given on the reference surface by

$$\zeta(s_0) = \begin{cases} \zeta^0 + \delta\zeta, & \text{if } s_0 \in [0, l_a] \\ \zeta^0, & \text{otherwise.} \end{cases} \tag{209}$$

One can verify that the spherical shape, given by $x(s) = R\sin(s/R)$, with $t_s^s = t_n^s = 0$, is a solution of the *Equations 137–142*. The strain has a jump $\delta u = u(l_a^+) - u(l_a^-) = \delta\zeta/(2K)$, such that

$$u(s) = \begin{cases} -(\zeta^0 + \delta\zeta)/(2K), & \text{if } s \in [0, l_a] \\ -\zeta^0/(2K), & \text{otherwise.} \end{cases} \tag{210}$$

Using this to solve *Equation 143* for $s \in [0, l_a']$ with $s_0(l_a') = l_a$ yields

$$\cos\frac{l_a}{R_0} - 1 = \left(\frac{R}{R_0}\right)^2 \frac{1}{1 + u(0)} \left(\cos\frac{l_a'}{R} - 1\right), \tag{211}$$

and similarly for $s \in [l_a', L]$

$$\cos\frac{l_a}{R_0} + 1 = \left(\frac{R}{R_0}\right)^2 \frac{1}{1 + u(L)} \left(\cos\frac{l_a'}{R} + 1\right). \tag{212}$$

These two equations determine the unknown radius $R$ and the deformed active region size $l_a'$ as

$$R = R_0\sqrt{\frac{1}{2}\cos\frac{l_a}{R_0}\left(u(L) - u(0)\right) + 1 + \frac{1}{2}\left(u(0) + u(L)\right)}, \tag{213}$$

$$l_a' = R\arccos\left(1 + \left(1 + u(0)\right)\left(\frac{R_0}{R}\right)^2\left(\cos\frac{l_a}{R_0} - 1\right)\right). \tag{214}$$

How the ratio $\frac{l_a'}{L}$ changes with the tension jump $\delta\zeta$ is plotted in *Appendix 8—figure 1* for $\zeta^0 = 0$. The above rescaling holds only for $\delta\zeta < 2K$ since the active region contracts to a point for $\delta\zeta \to 2K$.

For conserved volume, the spherical shape with $R = R_0$, $t_n^s = \partial_s t_s^s = 0$, $P = 2t_s^s/R$ is a solution of *Equations 137–142*. Here, only the relative size of the active patch changes from $l_a$ to $l_a'$. As before the strain is piecewise constant with a jump at $l_a$, $\delta u = u(L) - u(0) = \delta\zeta/(2K)$, and integrating *Equation 143* with $s_0(s = l_a') = l_a$, $s_0(s = L_0) = L_0$ gives the additional conditions

$$\cos\frac{l_a}{R_0} - 1 = \frac{1}{1 + u(0)}\left(\cos\frac{l_a'}{R_0} - 1\right), \tag{215}$$

$$\cos\frac{l_a}{R_0} + 1 = \frac{1}{1 + u(L)}\left(\cos\frac{l_a'}{R_0} + 1\right), \tag{216}$$

from which $u(0)$, $u(L)$, $l_a'$ can be determined.

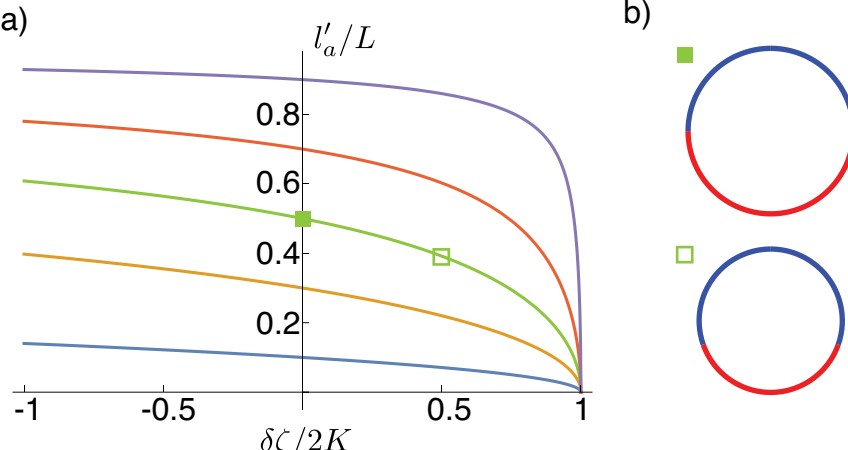

**Appendix 8—figure 1.** Rescaling of the size of the active region with active tension difference. (a) Plot of $l_a'/L$ as given by *Equations 213; 214*. This illustrates the rescaling of the active region size as a function of the tension difference $\delta\zeta$, for initial values $l_a/L_0 = 0.1, 0.3, 0.5, 0.7, 0.9$ (blue to purple). (b) Shapes corresponding to two points on the $l_a/L_0 = 0.5$ curve.

## Appendix 9

### Change of stability at a fold in a solution branch

We discuss here change of stability at a fold in a solution branch (*Maddocks, 1987*). We consider a dynamical system of the form $\partial_t \mathbf{x} = \mathbf{F}(\mathbf{x}, \lambda)$, with one control parameter $\lambda$. The line of steady state solutions is given by

$$\mathbf{F}(\mathbf{x}^*(s), \lambda(s)) = \mathbf{0} \tag{217}$$

where solutions are parametrised by $s$. Taking the derivative one obtains $(\partial_\mathbf{x} \mathbf{F})\,|_{\mathbf{x}^*}\, \partial_s \mathbf{x}^* + (\partial_\lambda \mathbf{F})\,|_{\mathbf{x}^*}\, \partial_s \lambda = \mathbf{0}$. We consider a fold in the solution curve where $\partial_s \lambda = 0$ and $|\partial_s \mathbf{x}^*| \neq 0$. In that case:

$$(\partial_\mathbf{x} \mathbf{F})\,|_{\mathbf{x}^*}\, \partial_s \mathbf{x}^* = \mathbf{0} \text{ at the fold,} \tag{218}$$

and the linear stability matrix at the fold $(\partial_\mathbf{x} \mathbf{F})\,|_{\mathbf{x}^*}$ thus has (at least) one eigenvalue that changes sign at the fold. Assuming that the solution branch is stable up to the fold, this indicates generically the appearance of an unstable mode; except in the special case where the 0 eigenvalue is a local maximum.

