## [Editor Report]

The article provides a physical description of shape transformations of epithelial tissues in three dimensions, subject to active forces generated within the cytoskeleton of the epithelial cells. The work is motivated by organoids and more generally by morphogenesis during development. Therefore, this study is useful not only for developmental biology but also for a general understanding of cellular properties, including membrane mechanics and cell shapes.

---

## [Decision Letter]

**Decision letter after peer review:**

Thank you for submitting your article "Active morphogenesis of patterned epithelial shells" for consideration by *eLife*. Your article has been reviewed by 3 peer reviewers, and the evaluation has been overseen by a Reviewing Editor and Naama Barkai as the Senior Editor. The following individuals involved in review of your submission have agreed to reveal their identity: Benoit Ladoux (Reviewer #1); Madan Rao (Reviewer #2); Jean-Francois Joanny (Reviewer #3).

All three reviewers positively evaluated the results of the study and congratulated the authors with a good achievement. According to one reviewer's opinion, the work is more appealing to a physics rather than biological audience and, therefore, could have a better home in a professional physics journal. This is, however a suggestion rather than a requirement.

The specific comments, which have to be addressed before the article can be recommended for publication are:

1. The authors consider the cases of contractile or extensile systems but biological systems can be composed of both particles. Could we find interesting properties by mixing contractile and extensile activities?

2. The spheroids are composed by passive and active parts determined by the ratio la/Lo. Is the passive part described with the exact same equations keeping active parameters at 0?

3. We could imagine that there are feedbacks between curvature and active stresses and bending moments with an evolution of the active parameters?

4. Dzhêta(cn) corresponds to the anisotropic active moment. It could come from various contributions either various activities at the basal versus the apical or various orders. I could imagine different configurations depending on these conditions. Could it be captured by the model? In particular, the authors could discuss the case where the nematic order on the apical versus is perpendicular to the one at the basal layer (as shown in the Figure 1). This may be important to understand the Hydra model where actin layers can be perpendicular to each other (See ref 22). Again, it would be interesting to imagine contractile and extensile layers at the basal and apical parts but may be beyond the scope of the current manuscript.

5. I had some difficulty to understand the shapes obtained at fixed volume for active bending moments (Figures5 d-e). The authors could better explain the indentations.

6. The concept of stress (tension) tensor and bending moment tensor has been used for sometime in the dynamics of curved passive membranes and active ones, see for instance, Sahu, A., Sauer, R. A., and Mandadapu, K. K., The irreversible thermodynamics of curved lipid membranes, Physical Review E, 96, 042409 (2017), and earlier papers by M. Lomholt.

7. Since this is a fluid description over scales larger than the cell size, the material density of the active agents will diffuse and flow to even out gradients. This surface diffusion is being ignored. I didn't see a separate equation for this local material density field, which I would have expected have surface diffusion and a curvature dependent surface advection. I presume this subsumed in the dynamics of the area density? This should be elaborated. The dynamics of the patterned active patch could be explained more with an accompanying schematic.

8. The trouble of taking the long length scale approach for the epithelial sheet context, is that this ignores the special physics of having individual cells, whose volume conservation needs to be preserved. This could result in changes in local thickness of the epithelium which contributes to additional elastic and dissipative stress components, and lead to other deformation modes.

9. No dissipative coupling to a dynamical fluid outside or inside. How is this justified?

10. Are there normal active stress components? The director field is in-plane. Again, it would help the reader to have all this stated, possibly pictorially.

11. Mechanical equilibrium solutions are obtained by setting the time derivatives to zero? The resulting equations will have many solutions, how is a unique one chosen. Does this have to do with Initial conditions. In this case, are these solutions stable? These points should be explained clearly.

12. Can the authors comment on the stability to nonaxisymmetric perturbations in this active dynamic context?

13. The dynamics show neck formation associated with finite time singularities. Can the authors extract a finite time singularity scaling form? Are there any predictions that can be made? This section could include a discussion on the relevance to endocytosis.

14. Could the authors comment on the existence of moving solutions? Ofcourse the authors don't have an external fluid or substrate onto which the cells can push on.

15. Since this manuscript is relevant for both active epithelial sheets and active membranes, the authors should refer to several earlier papers on active membranes such as Lomholt; Maitra et al; Mandadapu et al.

16. In principle the thin shell theory is a systematic expansion in powers of h/R_0 where h is the shell thickness and R_0 the radius of the undeformed tissue. At first order, one

obtains the tensions and at second order the bending moments. Because of that, some of the parameters of the theory are related to one another: there are only two independent viscosities and two independent active stresses.

This should be made more clear from the begining and the terms which are neglected should be of higher order in the expansion parameter. Is that the case?

17. The authors consider that the relaxation of the nematic order parameter is much faster than the relaxation of the shape. This is not obvious to me. Is there any reason to consider that the rotational viscosity is much smaller than the shear viscosity? If this is not true several terms must also be added in the force balance equations such as those related to the flow alignment parameter.

18. A parameter that is not discussed is the thickness of the cell layer. Is it supposed to be constant and if not, does its variation play any role? I guess that this is somewhat hidden in the parameter u. Is the cell volume supposed to be constant?

19. The constitutive equation 4 supposes that the longitudinal modes are elastic and the transverse modes are viscous. A more general theory would be to assume that the cell layer is a compressible liquid ie that the longitudinal modes are viscoelastic. Would that make any difference?

20.In equation 19, it seems to me that the coefficient \eta_V is proportional to the area of the cell layer and this can change significantly when the shape is deformed. Shouldn't that be taken into account?

---

## [Author Response]

All three reviewers positively evaluated the results of the study and congratulated the authors with a good achievement. According to one reviewer's opinion, the work is more appealing to a physics rather than biological audience and, therefore, could have a better home in a professional physics journal. This is, however a suggestion rather than a requirement.The specific comments, which have to be addressed before the article can be recommended for publication are:1. The authors consider the cases of contractile or extensile systems but biological systems can be composed of both particles. Could we find interesting properties by mixing contractile and extensile activities?

This is an interesting suggestion. For the case of isotropic active tension and isotropic active bending moments, the contractile and extensile cases are not intrinsically different, as only the difference in isotropic active tension or bending moments between the two regions matters, rather than their absolute values.

For the case of anisotropic active tension and anisotropic active bending moments, as the referee points out we have studied separately the case of a purely contractile and purely extensile cases, which lead to visibly different deformations (Figure 4a,f and 5a,d). To answer the referee question we have run examples of dynamic simulations with anisotropic active tension and anisotropic active bending moments, introducing a domain of contractile tension and the complementary domain of extensile tension:

**Author response image 1. sa2fig1:** Shapes obtained with two contiguous domains of extensile and contractile nematic active tension (left) or nematic active bending moments (right). Parameter values with + and – subscripts indicate values in the contractile and extensile regions. The contractile region is at the bottom of the shape.

We find that interesting shapes arise from this combination, which are roughly combinations of deformations observed for the pure extensile and pure contractile situation. We think this could be interesting to explore in a future study.

We also note that the active bending moment which we consider in the last part of the paper could arise from assembling two parallel layers with parallel nematic order, but one contractile and one extensile. We have now clarified this point (page 3, paragraph 3).

2. The spheroids are composed by passive and active parts determined by the ratio la/Lo. Is the passive part described with the exact same equations keeping active parameters at 0?

Yes that is the case, we now state it explicitly page 4, paragraph 3.

3. We could imagine that there are feedbacks between curvature and active stresses and bending moments with an evolution of the active parameters?

This is indeed a very interesting possibility that we mentioned in the discussion and that we now discuss more extensively (last paragraph). We note that a curvature-dependent active stress has been considered in the context of symmetry breaking of an active surface enclosing a passive fluid in Mietke et. al, PRL 2109, where it has no qualitative effect on the linear instabilities. Alternatively, one could consider a situation where the strength of active coefficients is a function of the trace or the determinant of the curvature tensor. Fully exploring this feedback could be the subject of future

4. Dzhêta(cn) corresponds to the anisotropic active moment. It could come from various contributions either various activities at the basal versus the apical or various orders. I could imagine different configurations depending on these conditions. Could it be captured by the model? In particular, the authors could discuss the case where the nematic order on the apical versus is perpendicular to the one at the basal layer (as shown in the Figure 1). This may be important to understand the Hydra model where actin layers can be perpendicular to each other (See ref 22). Again, it would be interesting to imagine contractile and extensile layers at the basal and apical parts but may be beyond the scope of the current manuscript.

These different cases are indeed taken into account in our model – in fact the case of orthogonal nematic order in the apical and basal layers, both giving rise to contractile stresses, as well as the case of equal nematic order in the apical and basal layers but with an extensile and contractile layer, precisely give rise to the anisotropic bending moment we consider in the manuscript. We also fully agree with the referee that our theory could describe the situation in the Hydra model which exhibits two parallel monolayers with perpendicular order. We have added a sentence on how the actin double-layer in Hydra could be represented in our model (page 3). We also have clarified in page 3 that the nematic active torque could arise from contractile layers with perpendicular order, or from a superposition of contractile and extensile layer, with nematic order with the same orientation.

5. I had some difficulty to understand the shapes obtained at fixed volume for active bending moments (Figures5 d-e). The authors could better explain the indentations.

We agree with the referee that the explanations in the text were not clear. We have rewritten this paragraph and we hope this is now more clearly explained.

6. The concept of stress (tension) tensor and bending moment tensor has been used for sometime in the dynamics of curved passive membranes and active ones, see for instance, Sahu, A., Sauer, R. A., and Mandadapu, K. K., The irreversible thermodynamics of curved lipid membranes, Physical Review E, 96, 042409 (2017), and earlier papers by M. Lomholt.

We thank the referee for pointing out these references to us, we have now added them to the manuscript (third paragraph of the introduction).

7. Since this is a fluid description over scales larger than the cell size, the material density of the active agents will diffuse and flow to even out gradients. This surface diffusion is being ignored. I didn't see a separate equation for this local material density field, which I would have expected have surface diffusion and a curvature dependent surface advection. I presume this subsumed in the dynamics of the area density? This should be elaborated. The dynamics of the patterned active patch could be explained more with an accompanying schematic.

We agree with the referee that in general, active agents could diffuse to even out gradients. In our work, we consider a situation where this effect is negligible and assumed that the active region is simply advected by the flow on the surface. This could arise for instance, from domains of gene expression in an epithelium, regulating differently the cytoskeleton contractility. In the discussion, we discuss an extension of the model taking into account chemicals regulating the activity. Following the request of the referee, we have added a schematic in Figure 1b to clarify that in our description, the active region is simply advected by the flow.

8. The trouble of taking the long length scale approach for the epithelial sheet context, is that this ignores the special physics of having individual cells, whose volume conservation needs to be preserved. This could result in changes in local thickness of the epithelium which contributes to additional elastic and dissipative stress components, and lead to other deformation modes.

We agree with the referee that this is a really interesting aspect of epithelial mechanics– although not necessarily in contradiction with considering long length scales, as one could introduce a tissue height field which varies slowly in space. We now discuss this point in more details in the discussion.

9. No dissipative coupling to a dynamical fluid outside or inside. How is this justified?

In the context of epithelial tissue embedded in water, the viscosity of an epithelium is 6 or 7 order of magnitude larger than the viscosity of the fluid medium around, making this approximation highly justified. We now state this point explicitly after Equation 4. We note that this consideration also applies when considering flows of the cell cortex.

10. Are there normal active stress components? The director field is in-plane. Again, it would help the reader to have all this stated, possibly pictorially.

By normal stress component, one could refer to tni, the normal part of the tension tensor ti. This component can not have an independent active contribution as it is fully constrained by the tangential torque balance, Equation 3. One could also refer to a normal contractile stress acting to decrease the height of the epithelial shell. Such a component can not be present in our theory, as we are not describing explicitly the height of the tissue. However, one could argue that with volume conservation, such a stress would result effectively in an isotropic tangential active stress, which we are considering (Equation 5).

We agree that we were not fully explicit about the order being tangent to the surface; this is now corrected (see page 2, paragraph 5). We are hoping that Figure 1c provides with a schematic which clarifies that we are considering tangent nematic order.

11. Mechanical equilibrium solutions are obtained by setting the time derivatives to zero? The resulting equations will have many solutions, how is a unique one chosen. Does this have to do with Initial conditions. In this case, are these solutions stable? These points should be explained clearly.

Mechanical equilibrium equations are indeed obtained by considering the force balance equations with a vanishing velocity field, as described in Appendix F. A solution branch is then found by increasing the active tension or bending moment difference, starting from a spherical solution where the system is entirely passive (Appendix F2). We now clarify this in section IIC. Although the solution branch is sometimes lost through a fold, we did not find any additional solution branches beyond the bifurcation point using our dynamical set-up.

Regarding solution stability, we give arguments in Appendix I showing that a fold in a solution branch generally signals a loss of stability; therefore we expect steady-state solutions to be stable up to a fold.

To illustrate numerically the stability of the steady states, we have plotted the results of dynamics simulations that relax to steady state into the corresponding pressure vs. control parameter or volume vs. control parameter diagrams, respectively, for several control parameter values per diagram:

**Author response image 2. sa2fig2:** Results of dynamics simulations that converge to a steady state (light blue circles) overlaid with solution branches from the main text. Since in the dynamics a smooth profile is used, rather than a step function as in the direct calculation, there is a small deviation between the two in some cases. We checked on two cases that if the smooth profile is used in the steady-state equation the match becomes excellent (b, blue dotted lines).

The results of dynamic relaxation fall exactly onto the solution branches. In particular, in cases that exhibit a fold in the solution branch, this shows that up to that fold the branch corresponds to stable solutions.

12. Can the authors comment on the stability to nonaxisymmetric perturbations in this active dynamic context?

We agree that this is an important question, but one which we cannot easily answer with our current analytical or numerical set-up. Extending our analysis to nonaxisymmetric shapes would be a significant effort which we postpone to future work.

13. The dynamics show neck formation associated with finite time singularities. Can the authors extract a finite time singularity scaling form? Are there any predictions that can be made? This section could include a discussion on the relevance to endocytosis.

This is an excellent question which prompted us to reexamine our statement on constriction of the neck in finite time. In fact, we noticed that the late time behaviour for very small neck size depends on the choice of numerical tolerance set for the solver. Rerunning our simulations with more stringent tolerances revealed that the constriction of the neck observed in Figure 2d seems consistent with exponential relaxation in time, and so does not occur in finite time. We have revised the manuscript accordingly and removed the reference to constriction occurring in finite time.

14. Could the authors comment on the existence of moving solutions? Ofcourse the authors don't have an external fluid or substrate onto which the cells can push on.

In the situations we have considered in our manuscript, we never observe a steady-state tangential flow on the surface, and so we would not have permanently moving solutions if the resistance of the external fluid was taken into account. This observation is probably a consequence of the assumption of bulk elasticity, which is relevant to the behaviour of epithelial tissues without cell division and death.

15. Since this manuscript is relevant for both active epithelial sheets and active membranes, the authors should refer to several earlier papers on active membranes such as Lomholt; Maitra et al; Mandadapu et al.

We thank the referee for pointing out these manuscripts and we now cite them in introduction.

16. In principle the thin shell theory is a systematic expansion in powers of h/R_0 where h is the shell thickness and R_0 the radius of the undeformed tissue. At first order, oneobtains the tensions and at second order the bending moments. Because of that, some of the parameters of the theory are related to one another: there are only two independent viscosities and two independent active stresses.This should be made more clear from the begining and the terms which are neglected should be of higher order in the expansion parameter. Is that the case?

We thank the referee for this interesting comment. Another way to state that constitutive equations arise from an expansion of h/R_0, is to say that they arise from an expansion in the curvature tensor. Equations 7 and 8 do indeed arise from an expansion in the curvature and nematic tensor (page 3, top paragraph: “We note that the equations 7-8 can be seen as generic constitutive equations for a nematic active surface with broken up-down symmetry but no broken chiral or planar-chiral symmetry, arising from an expansion in the curvature tensor and in the nematic order parameter Q_{ij} ») In this study, we only retain terms of order 0 in the curvature in the tension tensor, and expand the passive part of the bending moment tensor to order 1 in the curvature, giving rise to a bending modulus. The reason for keeping terms up to order 1 in the curvature specifically in the bending moment tensor, is that the bending modulus is required to avoid unphysical cusps.

In a recent preprint, one of us and collaborators have obtained a more general expansion of nematic surfaces active tensions and bending moments (see “Theory of nematic and polar active fluid surfaces”, G. Salbreux, F. Jülicher, J. Prost, A. Callan-Jones, 2022, PRR). We have now added a sentence after Equations 7 and 8 to refer to the manuscript mentioned above for a more general list of possible couplings.

The way that parameters of the theory are related to one another through this expansion strongly depends on the more microscopic model for the thin shell. For a homogeneous thin shell, the parameters of the theory have clear relationships (for instance, for an elastic shell of 3D elastic modulus E and thickness h, the 2D elastic modulus is Eh and the bending modulus Eh^2/12). However, we argue that more complex materials, for instance arising from two parallel active layers, do not necessarily have such constraints.

17. The authors consider that the relaxation of the nematic order parameter is much faster than the relaxation of the shape. This is not obvious to me. Is there any reason to consider that the rotational viscosity is much smaller than the shear viscosity? If this is not true several terms must also be added in the force balance equations such as those related to the flow alignment parameter.

We agree that this would be the case for a “standard” liquid crystal. However, we are considering a theory here to describe epithelia where the order parameter could react on a very different timescale than the tissue flow. For instance, the order parameter could be set by a cell planar polarity system which would be determined by the dynamics of a signalling pathway, intrinsically different from the timescale of mechanical stresses that set the flows.

We agree that this is a limitation of the current study and it would be interesting to later study the dynamics of the order parameter, rather than assuming that it relaxes quickly. We have now added a paragraph on this point in the discussion.

18. A parameter that is not discussed is the thickness of the cell layer. Is it supposed to be constant and if not, does its variation play any role? I guess that this is somewhat hidden in the parameter u. Is the cell volume supposed to be constant?

The referee is right that the parameter u, a 2D area elasticity, might actually arise, at least in part, from 3D volume conservation coupled to cell height elasticity. The simplified representation that we use, where the material simply has a 2D area elasticity, does not make a statement about whether the volume is conserved or not. We have added a paragraph in the discussion to clarify this point, and state that an explicit description of the tissue height is something that could be considered in the future.

19. The constitutive equation 4 supposes that the longitudinal modes are elastic and the transverse modes are viscous. A more general theory would be to assume that the cell layer is a compressible liquid ie that the longitudinal modes are viscoelastic. Would that make any difference?

This would indeed make a very big difference, as this would allow for steady-state tangential flows, something that we do not observe in the current theory. However, such a situation for a tissue would require cell division and death to occur, in order to relax two-dimensional bulk elastic stresses; something which is outside of the scope of our study. This point is now mentioned in the discussion.

20. In equation 19, it seems to me that the coefficient \eta_V is proportional to the area of the cell layer and this can change significantly when the shape is deformed. Shouldn't that be taken into account?

We agree that there is a potentially interesting physics to discuss here; however we introduced the coefficient \eta_V rather for numerical stability and in practice set up a small value for this coefficient; so that we are really studying the case with P=0. We have now clarified this in the text.